# Orographic resolution driving the improvements associated with horizontal resolution increase in the Northern Hemisphere winter mid-latitudes

Paolo Davini[1], Federico Fabiano[2], and Irina Sandu[3]

[1]Consiglio Nazionale delle Ricerche, Istituto di Scienze dell'Atmosfera e del Clima (CNR-ISAC), Torino, Italy
[2]Consiglio Nazionale delle Ricerche, Istituto di Scienze dell'Atmosfera e del Clima (CNR-ISAC), Bologna, Italy
[3]European Centre for Medium-Range Weather Forecasts (ECMWF), Reading, UK

**Correspondence:** Paolo Davini (p.davini@isac.cnr.it)

**Abstract.** In recent years much attention has been devoted to the investigation of the impact of increasing the horizontal resolution of global climate models. In the present work, a set of atmosphere-only idealised sensitivity simulations with EC-Earth3 have been designed to disentangle the relative roles of increasing the resolution of the resolved orography and of the atmospheric grid. Focusing on the winter Northern Hemisphere, it is shown that if the grid is refined while keeping the resolved orography unchanged, model biases are reduced only in some specific occasions. Conversely, increasing the resolved (or mean) orography is found to clearly reduce several important systematic model errors, including synoptic transient eddies, the North Atlantic jet stream variability and atmospheric blocking frequency and duration. From an analysis of the radiation budget it is concluded that the large changes in radiative fluxes caused by the resolution increase - something commonly observed in climate models - have a relevant impact on the atmospheric circulation, partially offsetting the benefits obtained from the increase in orographic resolution. These findings point to the necessity of always tuning climate models to fully exploit the benefits of high horizontal resolution.

## 1 Introduction

Global climate models (GCMs) have been shown to be powerful tools for understanding Earth's climate variability and for estimating its future evolution. Consequently, a considerable effort has been undertaken in the last decades to improve their reliability and accuracy. The continuous scientific development, which has resulted in improved parameterization schemes and novel numerical techniques, combined with supercomputing technologies and capabilities have allowed for notable steps forward in both the quality of the simulated present-day climate and in the reliability of future projections. The increased computational power has for example allowed to: 1) include additional components to better represent the Earth System (e.g. Flato, 2011) 2) increase the ensemble size to better sample unforced variability (e.g. Kay et al., 2015; Wyser et al., 2021) 3) increase model horizontal and vertical resolution to increase the fidelity of climate simulations by more explicitly representing key processes for the atmospheric or oceanic circulations (e.g. Haarsma et al., 2016).

This latter point has been explored by several modelling initiatives, such as the Athena (Jung et al., 2012), the UPSCALE (Mizielinski et al., 2014) or the Climate SPHINX (Davini et al., 2017b) projects, which all aimed at exploring the benefits of an increased horizontal resolution. More recently, the community-wide effort made in the framework of the HighresMIP project (Haarsma et al., 2016) further demonstrated the capability of high resolution models of improving several characteristics of the mean climate and its variability (e.g. Roberts et al., 2020; Fabiano et al., 2020; Bellucci et al., 2021; Zhang et al., 2021). More generally, recent outcomes from the Coupled Model Intercomparison Project Phase 6 (CMIP6, Eyring et al., 2016) also highlighted that - on average - higher resolution GCMs outperform lower resolution ones (e.g. Davini and d'Andrea, 2020; Fabiano et al., 2021; Priestley et al., 2020; Schiemann et al., 2020).

The model effective resolution, i.e. the smallest scale reasonably represented by a numerical model, is considerably larger than the model mesh. Sampling theory already predicts that two grid increments are required to correctly represent data, but due to numerical diffusion, aliasing and anti-aliasing filters the model effective resolution is estimated to be between three and five times the grid spacing (Klaver et al., 2020). Since increasing horizontal resolution implies smaller numerical truncation errors when solving the equations of motion, a finer grid positively affects the dynamics, thus leading to better resolved atmospheric eddies at a finer scale - as can be seen for tropical cyclones (Roberts et al., 2015; Vidale et al., 2021).

However, while the dynamical aspects of a GCM certainly take advantage of a finer grid, numerical schemes and physical parameterizations might respond in a less coherent way to a resolution increase: many aspects of radiation budget and hydrological cycle are also affected by model horizontal resolution (Vannière et al., 2019; Bador et al., 2020). For instance, timesteps are decreased when moving to higher resolution to accommodate numerical instabilities, implying indirect changes in convective adjustment timescales (Nordeng, 1994) or in the radiation scheme (Hogan and Hirahara, 2016). Furthermore, aspects of the dynamical core such as the advection scheme might not be perfectly conserving so that unexpected sources or sinks of heat may be present (Lucarini and Ragone, 2011; Mauritsen et al., 2012; Hobbs et al., 2016): the magnitude of these errors might be sensitive to the model grid (Polichtchouk et al., 2019).

One often neglected aspect is that studies which investigate the role of horizontal resolution do not perform re-tuning of the model at higher resolution (e.g. Haarsma et al., 2016). Climate model tuning is a fundamental aspect of model development, which relies on adjusting parameters from sub-grid parameterizations (usually involving clouds and convection) aiming at reducing model systematic errors (Hourdin et al., 2017). It is therefore hard to disentangle the contribution from the tuned/untuned parameterizations and the actual grid refinement: in the few circumstances when this has been done, the role of tuning has been shown to be non-negligible for several radiative and hydrological fields (Terai et al., 2018).

Another model feature which is significantly affected by horizontal resolution is the level of detail at which the model orography is resolved. Differently to convection which only becomes resolved at km-scale, orographic effects are partially resolved and partially parametrized from hundreds of km to km-scale resolutions, due to the fact that the orographic effects encompass a variety of scales across this resolution range. Orography is well-known to considerably impact the Northern Hemisphere winter circulation from daily to climate timescales (Held et al., 2002; Sandu et al., 2016; Pithan et al., 2016; van Niekerk et al., 2017). This is due both to the direct forcing on the planetary waves induced by the large-scale orographic barriers such as Rocky Mountains or Tibetan Plateau (Brayshaw et al., 2009; White et al., 2021) and to the small-scale processes,

such as turbulent orographic form drag, blocking of the flow at low levels and breaking of orographically generated gravity waves in the upper-troposphere and stratosphere (Lott and Miller, 1997; Beljaars et al., 2004; Sandu et al., 2019). However, the representation of orographic processes in weather and climate models remains to date uncertain, in particular due to the difficulties in constraining orographic drag processes and the parameterizations used to represent them (Sandu et al., 2019). Thus, many questions related to orographic processes and their impacts on the atmospheric flow remain open.

In this study, one of these aspects is explored: to what extent the increased fidelity of the higher horizontal resolution climate simulations is due to the fact that more orographic effects become explicitly resolved. Following Kanehama et al. (2019), who explored this question for weather timescales, hereafter the term "atmospheric resolution" will be used to refer to the grid spacing at which the atmosphere and land surface are discretized. Similarly, the term "orographic resolution" will be used to refer to the resolution of the grid box mean (or resolved) orography. This allows to create a clear distinction between the model grid spacing and the level of details of the orography.

Kanehama et al. (2019) demonstrated that - in the absence of orographic drag parameterizations - increases in orographic resolution are responsible for most of the improvement in the Northern Hemisphere winter medium-range forecast skill obtained when increasing the horizontal resolution of a numerical weather prediction system. Previous evidence also suggests that at climate timescales the orographic resolution may be responsible for an important part of the benefits of the increase in the horizontal resolution (Jung et al., 2012; Berckmans et al., 2013). Increasing the resolution of the resolved orography increase the level of details at which mountains are represented within the model: for example, the maximum height of the orographic barriers is elevated, so that to orographic-induced drag is larger and zonal winds are slowed down (White et al., 2021). All these changes, which are not directly dependent from the model atmospheric resolution, might have a large impact on several aspects of the mid-latitude climate variability (e.g. jet stream dynamics, atmospheric blocking, transient eddies activity): those aspects have never been explored in a comprehensive way. Given the considerable efforts to increase horizontal resolutions of GCMs in recent years, it is important to assess in a more quantitative manner to what extent the orographic resolution increase is responsible for the improvements obtained when increasing the horizontal resolution of the atmospheric component of GCMs.

This study thus aims to investigate the relative roles of the orographic and atmospheric resolution in shaping the simulated climate, extending and deepening the analysis developed at weather timescale by Kanehama et al. (2019) to a climate framework. It will thus try to shed light on the impact on both the mean climate and its variability, with a special focus on the winter Northern Hemisphere mid-latitude dynamics. This is done with a set of idealised atmosphere-only simulations carried out with the EC-Earth3 GCM at three different horizontal resolutions (from ∼80 to ∼25 km).

Section 2 will include the presentation of the experimental setup and of the diagnostics used to investigate the mid-latitude climate. Section 3 and Section 4 will respectively include the analysis of the mean climate and of the mid-latitude variability. Finally, Section 5 will investigate the role of the radiative budget in the framework of the results found, and Section 6 will present the final discussion and conclusions.

## 2    Data and Methods

### 2.1    Experimental setup

A set of sensitivity experiments has been performed with the atmosphere-only configuration of the EC-Earth3 Earth System Model (Döscher et al., 2021). The atmospheric component of EC-Earth3 is based on the Integrated Forecast System (IFS) cy36r4 developed by the European Centre for Medium Range Weather Forecast (ECMWF). The CMIP6 configuration of the model has been used, with the default CMIP6 resolution (TL255, which corresponds roughly to a 80-km grid spacing) and with two higher horizontal resolutions (TL511 and TL799, about 40 and 25 km respectively). The vertical resolution, which consists of 91 vertical levels (with the model top at 1 Pa), is kept the same in all simulations.

All the sensitivity experiments are atmosphere-only. In order to reduce as much as possible the possible sources of forced variability, Sea Surface Temperatures (SST) and Sea-Ice Concentrations (SIC) boundary conditions are provided as climatological cycles derived from the years 1985-2015 of the input4MIPs data (Durack et al., 2018). Similarly, greenhouse gases, ozone and aerosol concentrations are fixed to the year 2000 of the CMIP6 historical forcing (Eyring et al., 2016). Each integration lasts 31 years: 30 years are considered for the analysis, allowing for a one-year spinup. The TL799 experiments are performed for only 24 years due to their larger computational cost. However, considering only a 23-year time window for all experiments does not affect the results (not shown). It is important to point out that no extra tuning has been performed for the higher resolution configurations, while the low resolution TL255 configuration had been tuned prior to performing the EC-Earth3 CMIP6 simulations.

Orographic parameterizations of subgrid orographic processes are known to have a considerable impact on Northern Hemisphere circulation. Two parameterizations are used in the IFS, namely the Turbulent Orographic Form Drag (TOFD, Beljaars et al., 2004) and the Subgrid Scale Orography (SSO, Lott and Miller, 1997). The TOFD scheme represents the form drag due to orographic features with horizontal scales less than 5 km, while the SSO scheme represents drag to orographic features with horizontal scales larger than 5 km, associated with gravity wave breaking and with low level flow blocking. Both schemes decelerate the zonal flow: however, since they depend on the strength of the zonal wind, they indirectly interact between each other (Sandu et al., 2016) and with the resolved orography (van Niekerk et al., 2018). Therefore, their interaction will be resolution dependent - since mean wind will be different in each configuration. Consequently, in order to disentangle the impact of the resolved orography on the winter Northern Hemisphere circulation, both the TOFD and SSO schemes are turned off in all the sensitivity experiments.

This follows what was done in Kanehama et al. (2019), in order to avoid interactions between resolved and unresolved orographic processes. As expected, turning off the two parameterizations leads to a considerable deterioration of the mean climate, causing an increase in the jet speed at both low and upper levels. This corroborates previous studies which have demonstrated the major role played by these schemes for a realistic representation of the Northern Hemisphere circulation (Palmer et al., 1986; Pithan et al., 2016; van Niekerk et al., 2017, 2018; Sandu et al., 2019; White et al., 2021).

In this setup, five experiments were performed:

- Three control runs at TL255, TL511 and TL799 (labelled *ctrl* hereafter).

- Two runs at TL511 and TL799 with the mean orography at TL255 resolution (labelled *orog255* hereafter).

For the TL255-ctrl configuration, in order to have an estimate of the internal variability over the 30-year period, three ensemble members starting from different initial conditions have been run. The differences between the three members have shown to be generally small compared to the differences between TL255-ctrl and the other experiments (not shown). In order to reduce the atmospheric noise as much as possible, when analysing the TL255-ctrl the ensemble mean of the three integrations is used. Finally, one additional run at TL255 with the TOFD and SSO schemes active (labelled *ctrl-param* hereafter) has been performed. This is the same configuration as the one used for the CMIP6 integrations (Döscher et al., 2021).

Although a proper comparison with reanalysis datasets is not possible, due to the idealised character of the experiments, the ECMWF ERA5 Reanalysis (Hersbach et al., 2020) has been used as a reference to estimate the model biases. The time window considered is 1986-2015, which covers the window used for the SST/SIC forcing.

Multiple physical and dynamical fields, with both monthly and daily frequency, have been analysed focusing on the extended winter season from December to March (DJFM). Only radiative budgets have been estimated on the yearly timescale.

In Section 6, a brief analysis of the atmosphere-only simulations from a set of GCMs participating to the HighResMIP project (Haarsma et al., 2016) is carried out: data from CNRM-CM6 (Voldoire et al., 2019), EC-Earth3P (Haarsma et al., 2020), ECMWF-IFS (Roberts et al., 2018), HadGEM3-GC31 (Williams et al., 2018), IPSL-CM6A (Boucher et al., 2020) and MPI-ESM1-2 (Gutjahr et al., 2019) is used. All models have nominal resolutions ranging from 250 to 25 km, and for each model at least two versions are available, one at standard resolution and one at higher resolution; some models provide additional intermediate resolutions as well. All the models have been tuned in their low resolution configuration version, and the high resolution version is obtained by just increasing the grid space horizontal resolution, with no specific tuning.

In order to compare outputs of simulations performed at different resolutions, all data are interpolated on a common 2.5°x 2.5° grid with a bilinear remapping method.

## 2.2 Derivation of atmospheric and orographic resolution impacts

In order to summarise the discussion regarding the impact of the atmospheric and orographic resolution increases, a compact presentation of the five experiments has been adopted in several of the figures presented in Sections 3 and 4.

By comparing the *orog255* experiments at different resolutions (TL511 and TL799) with TL255-ctrl it is possible to estimate the net impact of the increase in atmospheric resolution while keeping constant the mean resolved orography. The impact of the "atmospheric resolution increase" is thus computed as the average of the differences between the TL799-orog255 and TL255-ctrl experiments and between the TL511-orog255 and TL255-ctrl experiments.

Similarly, by comparing the *ctrl* and *orog255* experiments (at both TL511 and TL799) it is possible to estimate the direct contribution of the better resolved mean orography. The impact of the "orographic resolution increase" is thus computed as the average of the differences between the TL799-ctrl and TL799-orog255 experiments and between the TL511-ctrl and TL511-orog255 experiments.

As expected, the signals obtained for the TL799 experiments are on average larger than the ones obtained for the TL511 experiments, but for most fields the two responses are consistent. Averaging the responses at TL511 and TL799 overcomes eventual issues arising from the limited length of the simulations by providing a more robust statistical sample.

## 2.3 Climate Variability Diagnostics

In order to assess the impact of the atmospheric and orographic resolutions on the winter Northern Hemisphere circulation different diagnostics are used. These metrics, which focus on several aspects of the synoptic-scale climate variability, are presented in the following paragraphs.

### 2.3.1 Jet Latitude Index

The variability of the North Atlantic eddy-driven jet stream is estimated through the Jet Latitude Index (JLI) developed by Woollings et al. (2010). The index describes the daily position of the low level jet over the Atlantic ocean, and it is defined as the daily latitude of the maximum of the zonal wind at 850hPa between 15°N and 75°N, zonally averaged between 60°W and 0°. In order to filter out high frequency variability, a 10-day Lanczos filter with a 31-day bandwidth is used.

### 2.3.2 Atmospheric Blocking Indices

Atmospheric blocking is a recurrent weather pattern typically occurring in the Northern Hemisphere at the exit of the Atlantic and Pacific jet stream (Tibaldi and Molteni, 1990; Davini et al., 2012), whose accurate simulation still represents a serious challenge for state-of-the-art GCMs (e.g. Davini and d'Andrea, 2020; Schiemann et al., 2020). In order to objectively recognize blocking events, several blocking indices have been developed (see Woollings et al., 2018). Those indices can be approximately clustered in mainly two families: those based on the reversal of some absolute field (e.g. Masato et al., 2011) and those based on the anomaly of some field exceeding a defined threshold (e.g. Schwierz et al., 2004). In the present work two indices, both based on the geopotential height at 500hPa (Z500) - one from each category - has been adopted.

Most of the analysis is carried out with a 2-D index based on the reversal of the meridional gradient of geopotential height, as done by Davini et al. (2012) (REV index hereafter). Two meridional geopotential height gradients at a southern (GHGS) and northern (GHGN) latitudes are defined:

$$GHGS(\lambda_0, \phi_0) = \frac{Z500(\lambda_0, \phi_0) - Z500(\lambda_0, \phi_S)}{\phi_0 - \phi_S}, \tag{1}$$

$$GHGN(\lambda_0, \phi_0) = \frac{Z500(\lambda_0, \phi_N) - Z500(\lambda_0, \phi_0)}{\phi_N - \phi_0} \tag{2}$$

and $\phi_0$ ranges from 30°N to 75°N while $\lambda_0$ ranges from 0° to 360°. $\phi_S = \phi_0 - 15°$, $\phi_N = \phi_0 + 15°$. Instantaneous Blocking for the reversal index (REV hereafter) is thus identified when:

$$GHGS(\lambda_0, \phi_0) > 0 \qquad GHGN(\lambda_0, \phi_0) < -10 \text{ m/°lat} \tag{3}$$

A second 2-D index based on the geopotential height anomaly from the mean flow has been computed adapting the definition given by Schwierz et al. (2004), as also done by Woollings et al. (2018) (ANO index hereafter). Daily Z500 anomalies are computed for each grid point as the difference with respect to the climatological mean for each dataset. Instantaneous blocking is detected as areas where daily Z500 anomalies exceed the 90th percentile of the Z500 anomaly distribution over 50°-80°N.

For both ANO and REV indices further spatial and temporal constraints are introduced: this ensures that blocking covers a sufficient area and persists for at least 5-day. Those constraints are applied according to Davini et al. (2012), thus defining the Blocking Events. The percentage of days per season in which Blocking Events occur (i.e., the number of blocked days) defines the blocking frequency climatology. Similarly, it is possible to define the Blocking Events duration as the average persistence of Blocking Events for each grid point. A complete description of the REV blocking climatology and of the blocking detection scheme and of its caveats and benefits may be found in Davini et al. (2012).

### 2.3.3 High frequency variability

High-frequency variability is key to investigating the behaviour of transient eddies at mid-latitudes. Here it is measured by applying a bandpass Fourier filtering between 2 and 6 days: filtered variables are hereafter indicated with a prime. Transient eddy activity is evaluated by using the standard deviation of the bandpass filtered daily geopotential height at 500hPa. In analogy, the upper tropospheric transient eddy kinetic energy is computed using the bandpass filtered zonal and meridional wind.

### 2.3.4 Barotropic and baroclinic energy conversion

Two more diagnostics are used to evaluate the transfer of energy from the mean flow to the eddies and vice versa. The role of the eddy forcing on the large-scale flow is analysed using the scalar product $\boldsymbol{E} \cdot \boldsymbol{D}$, which is a measure of barotropic exchange of kinetic energy between the transient eddies and the large-scale flow (Cai and Mak, 1990). $\boldsymbol{E}$ is defined as the horizontal part of the local Eliassen-Palm vector (Trenberth, 1986) which is very similar to the Hoskins E-vector (Hoskins et al., 1983) and is computed evaluating the bandpass filtered $u$ and $v$ as :

$$\boldsymbol{E} = \left( \frac{v'^2 - u'^2}{2}, -u'v' \right). \tag{4}$$

Conversely, $\boldsymbol{D}$ is the deformation of the mean field, where $D_x$ is the stretching deformation and $D_y$ the shear deformation (Cai and Mak, 1990; Black and Dole, 2000).

$$\boldsymbol{D} = \left( \frac{\partial \overline{u}}{\partial x} - \frac{\partial \overline{v}}{\partial y}, \frac{\partial \overline{v}}{\partial x} + \frac{\partial \overline{u}}{\partial y} \right). \tag{5}$$

$\boldsymbol{E} \cdot \boldsymbol{D}$ is usually computed in the upper troposphere (i.e., 250 hPa) where it reaches its highest values. Positive values of the $\boldsymbol{E} \cdot \boldsymbol{D}$ scalar (also known as barotropic energy conversion) indicate regions where the mean flow is feeding the synoptic eddies, while negative values point to regions where the mean flow is fed by the eddies. Large negative values are commonly seen in the exit region of the storm track, where the eddies are "barotropizing" the flow while smaller positive values are found in the entrance region of the storm track (Black and Dole, 2000)

Similarly to the barotropic conversion, a baroclinic conversion energy term is introduced. It is defined following Riviere and Joly (2006) as

$$F = -\frac{1}{S} v'\theta' \frac{\partial\overline{\theta}}{\partial y}, \tag{6}$$

which is the product between the meridional potential temperature gradient and the high-frequency meridional heat fluxes divided by a static stability parameter $S$, here defined as

$$S = -\frac{R}{p_0}\left(\frac{p_0}{p}\right)^{\frac{c_v}{c_p}}\frac{\partial\overline{\theta}}{\partial p}, \tag{7}$$

where $p_0$ is the reference pressure (i.e. 1000 hPa), $c_v$ and $c_p$ are the atmospheric specific heat at constant volume and pressure respectively and $R$ is the gas constant for dry air. The baroclinic conversion term $F$, usually evaluated in the lower troposphere (i.e., 850hPa), is mainly characterised by positive values, highlighting areas where the available potential energy of the mean flow is transferred to the eddies, namely in the core of the storm tracks (Cai and Mak, 1990; Riviere and Joly, 2006).

### 2.4 Statistical significance and performance diagnostics

In order to assess whether the changes induced by orographic/atmospheric resolution changes are significant or not, a Welch t-test (at 95% or 99% significance level) has been used assuming independence between consecutive winters.

Finally, aiming at objectively measuring the model improvements associated with orographic or atmospheric resolution, two scalar diagnostics are extensively used in the manuscript: 1) the *pattern correlation improvement* (PCI) and 2) the *relative RMSE improvement* (RRI) and. PCI is defined as the area-weighted (or pressure-weighted) Pearson correlation coefficient between the TL255-ctrl systematic error (i.e. the TL255-ctrl minus ERA5 climatology) and the changes induced by the atmospheric or orographic resolution. RRI is the area-weighted (or pressure-weighted) relative RMSE change following the increase in orographic/atmospheric resolution compared to the RMSE of the TL255-ctrl against the ERA5 reanalysis. For both diagnostics, a negative number implies a resolution-induced change which reduces the model bias, while a number close to zero (or positive) might suggest an unmodified (or a deteriorated) model bias.

## 3 Mean climate impact

The impact of the atmospheric and orographic resolutions is firstly explored in terms of key characteristics of the mean climate during the Northern Hemisphere winter season (DJFM). The TL255-ctrl experiment shows moderate biases in zonally averaged temperature and wind, as shown by Figure 1. It should be noted that this experiment is missing the sub-grid orographic parameterizations, and hence these biases are larger than in the default EC-Earth3 configuration (not shown). The TL255-ctrl experiment is characterised by an overly intense temperature gradient at upper levels and by a too weak gradient at lower levels (Fig. 1a), which is reflected into a too strong jet stream, especially in the upper troposphere, in both the hemispheres (Fig. 1d). Additionally, the stratospheric polar vortex is overestimated by several m/s. The unequal distribution of meridional temperature gradients is a recurrent problem already seen in previous versions of EC-Earth (Davini et al., 2017a).

As shown by Fig. 1b,c and Fig. 1e,f, the impact of the atmospheric resolution increase is completely different from the impact of the orographic resolution increase. In the zonal average, the most impressive change is the cooling of the stratosphere following the atmospheric resolution increase (Fig. 1b), which is associated with an increase of the tropopause height, possibly caused by changes in gravity waves representation and by errors in vertical advection (Polichtchouk et al., 2019). The changes in the tropospheric jets are however limited, mainly showing a small deceleration of both the northern and southern hemisphere jets (Fig. 1e). Conversely, the orographic resolution increase has a small impact on the tropical stratospheric temperature, but drives a warming of the polar stratosphere, likely associated with a larger orographic wave activity propagating upward (Fig. 1c). The jet streams are deflected, showing a poleward displacement in the Southern Hemisphere and an equatorial displacement in the Northern Hemisphere (Fig. 1f), which likely depends on the specific structure of the continental landmass in the two hemispheres. However, this reduces the temperature and wind biases in both hemispheres, as confirmed by the PCI and RRI values.

The discussed changes in temperature and wind have a complex longitudinal structure which is illustrated in Figure 2, where the upper level streamfunction and the lower level temperature and zonal winds are shown. The streamfunction changes are generally larger in the Northern Hemisphere, for both the atmospheric and orographic resolution increases: this is expected considering that the larger orographic barriers are found in the Northern Hemisphere. However, the way in which the orographic and atmospheric resolution impact the flow is very different. While both signals project on a strengthening of the Atlantic ridge, they have an opposite response over Pacific and North America. This can be better appreciated by looking at the 850hPa temperature and zonal wind changes.

Indeed, the impact of the increased resolution of the mean orography generally follows the expected theory associated with vorticity conservation where the orographic barrier induces a deflection of the flow (Valdes and Hoskins, 1991; Brayshaw et al., 2009; White et al., 2017). The higher and more detailed structure of the Rocky Mountains and Tibetan Plateau induces a stationary wave pattern (Fig. 2c), associated with a warming on the windward side of the mountain chains (Fig. 2f) and decreasing the jet speed especially over the Atlantic sector: interestingly, the Pacific jet stream is deflected equatorward (Fig. 2i). Overall, this goes into the direction of reducing the systematic model errors seen for the TL255-ctrl (Fig. 2a,d,g). The same general result holds for the Southern Hemisphere (summertime) circulation, where the increased height of the Andes produces a poleward shift of the jet stream (Fig. 2i), partially compensating the bias of the TL255-ctrl experiment (Fig. 2g).

The impacts of the atmospheric resolution increase are more difficult to understand. It leads to a cooling of the continental landmass (Fig. 2e) and a moderate decrease of the jet streams in their exit regions (Fig. 2h), associated with a wavy response in the streamfunction which is characterised by a southwest-northeast oriented pattern over the North Pacific (Fig. 2b). However, the temperature, wind and streamfunction responses do not always reduce the biases seen for the TL255-ctrl experiment (Fig. 2a,d,g). Indeed, improvements are obtained over large parts of the North Atlantic and Eurasia, but the wave-2 pattern over the North Pacific and North America is almost out of phase with the TL255-ctrl systematic error. More generally, the complicated patterns of these responses show how hard it is to understand the origin of the changes induced by the increase of the atmospheric resolution.

## 4    Changes in mid-latitude variability

Given the wide impact on the mean climate, it is also interesting to analyse the consequences of the atmospheric and orographic resolution increases on the winter Northern Hemisphere mid-latitude climate variability. A key characteristic that is worth investigating is the North Atlantic jet stream variability, which can be assessed by examining the Jet Latitude Index (JLI), shown in Figure 3.

This shows the usual trimodal peak which is characterised by a more frequent central peak around 45°N, an equatorward peak around 35°N and a poleward one around 55°N. While the central peak is associated with the zonal flow over the North Atlantic basin, the equatorward and poleward peaks are associated with cyclonic (i.e. negative North Atlantic Oscillation) and anticyclonic Rossby wave breaking respectively (Woollings et al., 2010). A typical bias of climate models is to have a weak trimodality (e.g. Anstey et al., 2013; Kwon et al., 2018), usually associated with a too strong and poleward displaced jet, which favours jet pulsing as the dominant mode of variability rather than jet wobbling (Barnes and Polvani, 2013). As many other GCMs, also the TL255-ctrl run shows a preponderant central peak and underestimated frequencies for the equatorward and poleward peaks. As can be seen in Figure 3, the atmospheric resolution increase provides a moderate reduction of the JLI bias: it reduces the frequency of the central peak and it slightly increases that of the equatorward one. Increasing the orographic resolution provides a larger improvement: on top of similar changes (but with larger magnitude) for the equatorward and central peaks, it also increases the frequency of the poleward peak. As expected, the TL799-ctrl simulation is the most realistic, having a reduced overall bias (albeit a remaining overestimation of the JLI around 45°N central peak and an apparent spurious fourth peak at 50°N are still seen). Overall, Fig. 3 further shows how also in terms of Atlantic jet variability at least the half of the improvements associated with the increased horizontal resolution are due to the change in orographic resolution.

Another important element of the Northern Hemisphere mid-latitude circulation is atmospheric blocking. Euro-Atlantic blocking activity is partially associated with the Atlantic jet variability (Hannachi et al., 2012; Kwon et al., 2018), but especially over Central Europe is substantially independent from it (Davini et al., 2014; Madonna et al., 2017). Most interestingly, state-of-the-art GCMs typically underestimate the frequency of blocking events over both the Euro-Atlantic and North Pacific sectors (Davini and d'Andrea, 2020). Figure 4a - where the TL255-ctrl bias for the REV blocking index is shown - confirms this view, showing a massive underestimation of blocking frequencies over the Central European sector and a marked overestimation at low latitudes over the Azores. This is a typical configuration associated with a too zonal flow over the North Atlantic sector, where the anticyclonic wave breaking activity is constrained at lower latitudes by an overly strong waveguide. As expected this bias is exacerbated by the absence of the orographic wave parameterizations, which considerably alleviate the systematic error (blocking climatology for the TL255-ctrl run is shown in Figure SM1).

Both the atmospheric and orographic resolution increases lead to larger blocking frequencies, especially at high latitudes. Over the North Pacific sector the impact of mean orography provides a clearer benefit than the atmospheric resolution increase, but over the Euro-Atlantic sector - also considering that the region attaining a 95% statistical significant level are limited - the reduction of the bias driven by mean orography resolution is similar to the one of the atmospheric resolution. This is confirmed by the PCI and RRI figures, which points to a reduction of about 10% of the model RMSE in both cases.

An interesting feature to investigate is the blocking duration, which has been for a long time assumed to be related to the transient eddy forcing (e.g. Shutts, 1983). The increase in atmospheric resolution should help improve the representation of these aspects since the size of the eddies which are resolved at TL799 is about four times smaller than at TL255. However, as seen in Figure 4e, the change in atmospheric resolution has a negligible impact on the blocking duration. Conversely, increasing the mean orography leads to a clearer impact on the duration of blocking events with a widespread increase over the Euro-Atlantic sector. It is thus reasonable to assume that the longer blocking duration is the result of the eddy-mean flow interactions operating over the North Atlantic, which are influenced by the Atlantic jet structure produced by the orographic resolution increase over the Rocky Mountains.

Given that blocking is quite sensitive to the index adopted, Figure SM2 shows the same analysis carried out in Figure 4 making use of the ANO index. Despite a radically different climatology, the negative bias is still observed over the Euro-Atlantic sector. Interestingly, the orographic resolution increase produces a widespread but not significant increase in terms of blocking events frequency and blocking duration, while marginal changes are observed increasing the atmospheric resolution.

This analysis - even if it confirms the beneficial impact of increased orographic resolution with both the ANO and REV indices, especially on blocking duration - puts further evidence on the sensitivity to blocking index and to the requirement of long integration to assess robustly the impact of any model change on atmospheric blocking.

Given the above-discussed changes in blocking duration, it is interesting to analyse the transient eddy activity, which is shown in Figure 5a,b,c. The TL255-ctrl experiment has a large positive bias (Fig. 5a), having storm tracks extended too far downstream over Europe and North America, almost continuously developing over Asia, and slightly displaced poleward. This effect - which is often seen in climate models - is exacerbated in our TL255-ctrl simulation by the absence of the orographic parameterizations, which plays a notable role in slowing down the westerly mid-latitude flow (Pithan et al., 2016; White et al., 2021). Increasing the mean orography (Fig. 5c) results in a bias reduction which, albeit of smaller amplitude, mirrors the patterns of the positive bias in transient eddy activity found for the TL255-ctrl experiment (Fig. 5a). The increase of the atmospheric resolution (Fig. 5b) has a moderate impact on the transient eddy activity over the Pacific and shows a complex signal over the Atlantic. A small decrease of the bias is seen on the eastern side of the basin, but it does not extend downstream over the Eurasian continent. The transient eddy activity increases over the North American continent, a region where the model bias is already positive in the TL255-ctrl experiment. These results suggest that the increase in blocking duration (seen in Fig. 4f) might be associated with a weakening of the transient eddies forcing due to the increase in orographic resolution.

The changes to the transient eddies can be partially explained by examining the baroclinic energy conversion, which is a measure of the energy extracted from the mean meridional temperature gradients and transferred to baroclinic eddies. Its pattern roughly matches the meridional heat fluxes and the Eady growth rate, since it relies on the same terms (not shown). The TL255-ctrl experiment is characterised by an underestimation of the baroclinic conversion term in the two storm tracks (Fig. 5d), which is consistent with the underestimated meridional temperature gradient in the lower troposphere seen in Figure 2. Overall - although with moderated statistical significance - the increase in atmospheric resolution does not affect Pacific baroclinicity, but it increases it over the North American continent, in line with the storm track strengthening over that region

(Fig. 5e). The orographic resolution increase improves the baroclinic conversion term over the Pacific storm track, and partially also over the Atlantic (Fig. 5f).

The barotropic energy conversion term shows different TL255-ctrl bias in the two oceanic basins: over the Atlantic it is characterised by a dipole, with a "barotropization" of the mean flow by the eddies occurring too equatorward and too eastward (Fig. 5g). As mentioned in Section 2, this term describes where the jet stream is feeding the eddies (when it is positive) or where the mean flow is extracting energy from the eddies (when it is negative). The TL255-ctrl bias suggests that transient eddies favour a southward displacement of the final part of the Atlantic jet, leading to an incorrect tilt of the North Atlantic jet stream, which tends to be too strong and too zonal (as seen in Fig. 2h, which in turn affects the blocking frequency as seen in Fig. 4). While the increase in atmospheric resolution does not change the behaviour of the eddies in the Atlantic (it actually worsens the simulation of the eddy extraction of energy over the North American continent), the orographic resolution increase partially reduces the model bias, reducing the barotropic conversion at lower latitudes and increasing it at higher ones, displacing the Atlantic jet poleward and increasing its southwest-northeast tilt (Fig. 5h,i). Nonetheless, it should be remarked that few regions attain a 95% statistical significance.

A summary of the influence of resolution on eddy dynamics can be extracted looking at the PCI and RRI diagnostics of Fig. 5, which are considerably larger when increasing the orographic resolution (showing a reduction of the RMSE of about 20%) than when increasing the atmospheric resolution.

Globally, it can be stated that large-scale changes introduced by the increase in resolved orography modify the way eddies and the mean flow interact. Albeit the stationary wave pattern over the Atlantic is similarly modified when both the atmospheric and orographic resolution are increased (Fig. 2b,c), the high frequency variability reacts in a significantly different way. This further highlights how finer atmospheric resolution alone (without an increase in the orographic resolution) is not able to improve the representation of eddy dynamics.

## 5   Radiative Budget Offset

Overall, the analysis presented in the previous sections showed that while the increase in orographic resolution improves the representation of the Northern Hemisphere winter mean climate and variability, the effect of the increase in atmospheric resolution is more complex and provides contrasting results. This is summarised in Figure 6, where the radar chart shows the root mean squared error (RMSE) for a set of variables both for the entire globe during the Northern Hemisphere winter season (Fig. 6a) and for the mid latitudes (Fig. 6b). Here the RMSE has been normalised by the RMSE of the TL255-ctrl experiment, so that this experiment has an RMSE of 1 for each variable. The larger the distance from the centre of the chart, the larger the RMSE is: a perfect match with the ERA5 reanalysis (i.e. RMSE = 0) would fall in the centre of the chart. An estimate of the internal variability is provided by the three ensemble members for the TL255-ctrl. In addition to the previously discussed experiments, Figure 6 also shows the TL255-ctrl-param experiment, which outperforms all the other configurations, especially in the Northern Hemisphere mid-latitudes, as expected given the importance of orographic parameterizations in current GCMs

Figure 6 illustrates that best results - among the experiments without orographic parameterizations - are obtained for the TL799-ctrl experiment, which improves significantly for almost every variable analysed, with a reduction of the RMSE of about 30% (when compared to TL255-ctrl). Only the upper tropospheric air temperature shows a net worsening, associated with the cold stratospheric bias seen in Figure 1b. The TL511-ctrl lags behind the TL799-ctrl, but still shows evident improvement in both the global and in the Northern Hemisphere mid-latitude RMSE. Conversely, the changes in both the TL799-orog255 and TL511-orog255 with respect to the TL255-ctrl are much smaller in almost all the metrics considered. This suggests that most of the improvements seen in these circulation aspects, which also include specific dynamical and high-frequency measures as barotropic energy conversion, eddy kinetic energy or meridional heat fluxes, are driven by the increase in the orographic resolution rather than by a refinement of the atmospheric resolution.

One interesting result, highlighted by Figure 6, is that for several fields TL799-orog255 has a larger RMSE than TL511-orog255, and sometimes both are worse than TL255-ctrl. This is further reinforced by the analysis of the specific changes associated with atmospheric resolution increase in a few selected fields (Figure SM3): both PCI and RRI are always larger (implying larger improvement) for TL511-orog255 than for TL799-orog255, meaning that finer atmospheric grid deteriorates the climate simulation. Conversely, the impact of mean resolved orography is larger for TL799 than for TL511 (Figure SM4), suggesting that finer orographic resolution improves the integrations. Overall, this is partially counterintuitive since there is no dynamical argument for which an atmosphere simulated on a finer horizontal grid, which has smaller truncation errors and finer-scale resolved eddies, should lead to a deterioration of fundamental aspects of the atmospheric circulation.

These findings can be explained taking into consideration that - as commonly done in resolution comparisons (e.g. Haarsma et al., 2016) - tuning has been performed only for the TL255 configuration. Conversely, TL511 and TL799 have not been retuned. Indeed, one major target of model tuning is the model top of atmosphere (TOA) radiative balance: this is usually achieved modifying parameters associated with convection or cloud microphysics (Mauritsen et al., 2012; Hourdin et al., 2017; Döscher et al., 2021), so that tuning operation has an important impact on the hydrological cycle, on cloud dynamics and - consequently - on the radiative budget. It is thus possible that the lack of a proper model tuning at high resolution has a two-fold impact on the atmospheric circulation: 1) it could lead to changes in clouds distribution which might affect the meridional temperature distribution (and consequently also mean wind profiles) and 2) it could affect convection and precipitation leading to changes in tropical Rossby Wave source and propagation which can dynamically force the mid-latitude climate. Both changes will be reflected by a different radiative budget, especially in tropical regions.

Figure 7 shows the global yearly-averaged TOA radiation (the net fluxes, and the outgoing longwave and shortwave radiation), the atmospheric imbalance (i.e. generated by systematic biases in the energy and mass conservation of the model, which leads to heat sources and sinks within the atmosphere, e.g. Berrisford et al. 2011) and total cloud cover and precipitation in the different EC-Earth3 experiments. Simulations with different atmospheric resolution present notable differences in all the variables considered, as commonly seen when horizontal resolution is increased (Davini et al., 2017b; Vannière et al., 2019). Larger changes with respect to the TL255-ctrl experiment are seen for the TL799 than for the TL511 experiments, suggesting that the more the horizontal resolution is increased the more the radiative balance is modified. Changes in these metrics with respect to TL255-ctrl are seen also in the *orog255* integrations. These are however similar or slightly smaller to those seen for

the *ctrl* integrations, suggesting that they are mostly driven by the increase in atmospheric rather than in orographic resolution. Finally, as it may be expected, the activation of the orographic parameterizations has a negligible impact on these global averages (compare TL255-ctrl and TL255-ctrl-param).

Increasing the resolution from TL255 to TL511 leads to an increase of almost 1 W/m$^2$ in the net TOA radiation (Fig. 7a). However, since the atmospheric imbalance (Fig. 7d) increases by a larger value (about 1.2 W/m$^2$), this implies that the net surface fluxes are slightly reduced and a minor decrease in surface temperature is observed (not shown). The change in net radiation at TOA is mainly associated with a reduction of the shortwave radiation reflected back to space, which suggests an overall decrease in Earth's albedo (Fig. 7c). The decrease in shortwave outgoing radiation is only partially compensated by the increase in the outgoing longwave radiation (OLR, Fig. 7b). Considering that 1) OLR is mainly driven by characteristics of tropical convection and 2) OLR is not influenced at all by the orographic resolution (*ctrl* and *orog255* experiments at TL511 and TL799 show the same OLR for a given resolution), the changes in OLR are likely due to the dependence of the convection (and precipitation rate) on the atmospheric resolution. Indeed, these are two aspects strictly connected with the model tuning, since parameters such as the entrainment rate for organised convection or the precipitation conversion rate are typically adjusted in this process (Döscher et al., 2021, e.g.).

The total cloud cover decreases by 1% at TL511 and by 2% at TL799 compared to TL255-ctrl, suggesting that the reduced radiation reflected back to space is due to the decreased cloud amount (Fig. 7e). Similarly, significant changes are observed for precipitation, showing increased rainfall at both TL511 and TL799 compared to TL255-ctrl (Fig. 7f). For both cloud cover and precipitation, changes in the tropical region are larger (not shown). More generally, the changes seem to be quite linear, always being larger for the TL799 than the TL511 experiment.

A more detailed spatial analysis shows considerable changes in the tropical areas (Figure SM3 and SM4), characterised by a moderate decrease in cloudiness. However, the changes are very complex, being likely associated with a redistribution of convection along the Equator. Less precipitation and convection are seen in the higher resolution experiments over the Maritime Continent, while increased precipitation and convection are seen over Western Pacific, the Indian Ocean and the Amazon. A reduction of cloud cover is also seen over the Peruvian coast, suggesting less stratocumulus there.

It is possible to conclude that changes in tropical convection, clouds and precipitation, which can be diagnosed through the deterioration of the radiative budget (which shows larger radiative imbalance as long as the model grid is moved away from the tuned configuration) counteracts any potential improvements provided by the refinement of the atmospheric grid. However, when the model horizontal resolution is increased these mixed effects are counterbalanced by the orographic resolution changes, which are more effective at higher resolution, and thus provide a net improvement of the simulation.

## 6   Discussion and Conclusions

In this study, the role of the resolution of the mean orography in shaping the Northern Hemisphere winter mid-latitude flow has been investigated with a set of idealised sensitivity experiments carried out with the atmospheric component of the EC-Earth3 climate model. This has shown that (in the absence of orographic drag parameterizations) most of the benefits in the Northern

Hemisphere winter midlatitudes induced by the increase in the horizontal resolution are actually caused by the changes in the mean resolved orographic resolution. Such improvements are evident in almost all the variables analysed, from large-scale mean state up to high frequency variability. This corroborates the results for medium-range forecasts timescales (Kanehama et al., 2019).

Indeed, the orographic resolution impact is quite clear, and supports previous theoretical studies with simplified setups and models (e.g. Brayshaw et al., 2009). The large orographic barriers in the Northern Hemisphere interact with the flow, deviating and decelerating it, giving rise to weaker (and - over the Atlantic - more tilted) eddy-driven jet streams. Increasing the atmospheric resolution alone has mixed impacts so that it does not necessarily reduce systematic model biases, especially over the Pacific basin.

The set up of the presented experiments followed the typical approach of the initiatives aiming at assessing the impact of horizontal resolution increases in GCMs, as the PRIMAVERA H2020 project or the HighResMIP project: EC-Earth3 was tuned only once at the standard resolution (TL255) and no re-tuning was performed for different resolutions (TL511, TL799). However, increasing the atmospheric resolution caused large changes in the radiative budget at the TOA, suggesting that the lack of a proper re-tuning of the finer resolution configurations can be considered as the mechanism responsible for the minor

improvements observed when the atmospheric resolution is increased. Indeed, the sensitivity of physical parameterizations (as convection and microphysics schemes) to grid spacing and time-stepping might have serious consequences on the final simulated precipitation and cloud structure. These changes - which can be seen by altered radiative fluxes - might substantially impact meridional temperature gradients and Rossby waves source and propagation (which ultimately shape the structure of the mid-latitude jet streams), partially offsetting the benefits that high atmospheric resolution provides, as for example a better

representation of synoptic fronts and tropical cyclones. Indeed, current experiments showed that when going from TL255 to TL511 both atmospheric grid refinement and mean resolved orography have a positive impact, but that at TL799 - while orography provide further benefits - the lack of tuning offsets the potential improvements due to the atmospheric resolution increase.

Although in the present work the relevance of mean orography resolution has been demonstrated only for EC-Earth3,

given their fundamental dynamical character these findings are likely valid for other GCMs. Most importantly, not only the orography-related conclusion might be extended to other climate models. The analysis of the radiative budget of a selection of models from the HighResMIP project shows that similar significant changes in the radiative fluxes, as well as in cloud cover and precipitation, are seen when the horizontal resolution is increased (Vannière et al., 2019). A simple summary is shown in Fig. 8, where the same variables plotted in Figure 7 are presented for a set of the HighResMIP models. While EC-Earth3P

- the older EC-Earth3 version which took part to HighResMIP - shows the largest sensitivity in the atmospheric imbalance, all the GCMs show significant changes in the net TOA fluxes (between 0.5 to 2 W/m$^2$) and changes in total cloud cover and precipitation up to several percents (Moreno-Chamarro et al., 2022).

It is therefore likely that a significant part of the benefits of the atmospheric resolution increase in HighResMIP might have been lost due to changes in tropical convection, clouds and precipitation which can influence the mid-latitude circulation.

We therefore strongly encourage that further initiatives such as HighResMIP should be based on a tuned version of the high

resolution configuration, making sure that convection and cloud microphysics parameterizations are correctly constrained to produce reliable globally averaged radiative fluxes. This is clearly a costly exercise in terms of computational resources, but a basic tuning of the atmospheric-only configuration at the top-of-the-atmosphere radiative fluxes could easily produce a smaller imbalance in the radiative fluxes and a more correct representation of the thermal structure of the troposphere. This will eventually sum up with the robust positive effects of the orographic resolution increases.

Although the role of the sub-grid orographic parameterizations, namely the TOFD and SSO schemes, was not the focus of this study, these schemes are known to result in huge improvements of the winter mid-latitude flow: indeed, a TL255 experiment with orographic parameterizations outperforms a TL799 run in many aspects (Fig. 6) without altering the radiative budget (Fig. 7). It is therefore crucial to pursue the development of such orographic schemes, since they could provide relevant benefits to the representation of the climate mean state and variability at a negligible computational cost. In this direction, the authors plan to investigate in future work with further targeted experiments the impact of TOFD and SSO schemes, and of changes therein, in climate simulations.

*Data availability.* EC-Earth3 integrations are part of the REFORGE ECMWF special project and are accessible upon request to the authors.

*Author contributions.* PD and IS designed the experiments. PD performed the integrations, conducted most of the data analyses and wrote the paper. FF performed the analysis on the HighResMIP data. All the authors contributed to the discussion, and commented and organised the paper.

*Competing interests.* The authors declare that they have no conflict of interest.

*Acknowledgements.* PD thanks ECMWF for providing computing time in the framework of the special projects SPITDAV2. FF has been supported by the European Commission (grant no. PRIMAVERA 641727).

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

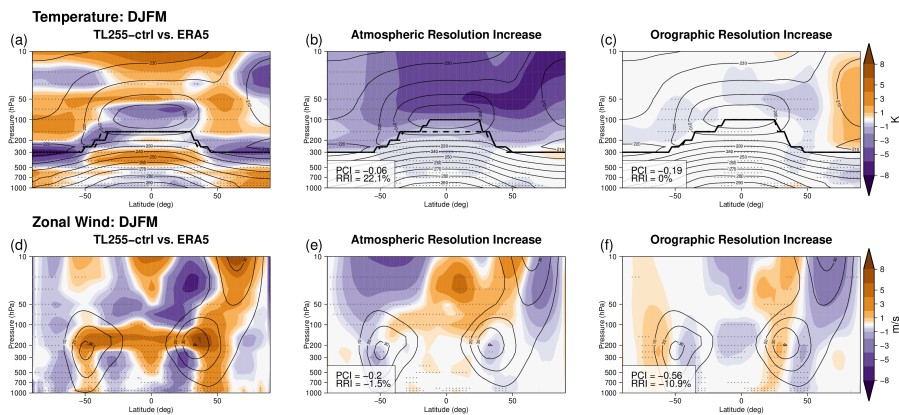

**Figure 1.** DJFM zonal mean temperature (upper row) and zonal wind (lower row): (a,d) EC-Earth3 TL255-ctrl bias with respect to ERA5, (b,e) changes induced by the atmospheric resolution increase and (c,f) changes induced by the orographic resolution increase. Shading shows differences, contours the TL255-ctrl field. Please note the irregular color-bar spacing. In (a,b,c) black lines show the tropopause height for reference (dashed) and changes (solid). Stippling indicates significance with a Welch t-test at the 1% level. In (b,c,e,f) the PCI and RRI (see text for details) are reported at the bottom left of each panel: large negative values implies reduced bias.

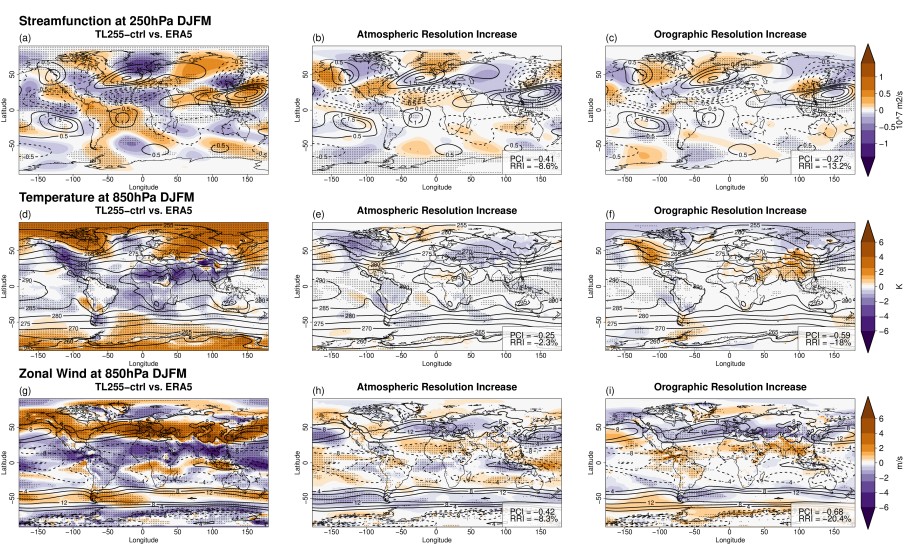

**Figure 2.** DJFM mean asymmetric component of the 250hPa streamfunction (top row), 850hPa air temperature (middle row) and 850hPa zonal wind (bottom row) for (a,d,g) EC-Earth3 TL255-ctrl bias with respect to ERA5 (b,e,h) changes induced by the atmospheric resolution increase and (c,f,i) changes induced by the orographic resolution increase. Shading shows differences, contours the TL255-ctrl field. Please note the irregular color-bar spacing. Stippling indicates significance with a Welch t-test at the 1% level. In (b,c,e,f,h,i) the PCI and RRI (see text for details) are reported at the bottom right of each panel: large negative values implies reduced bias.

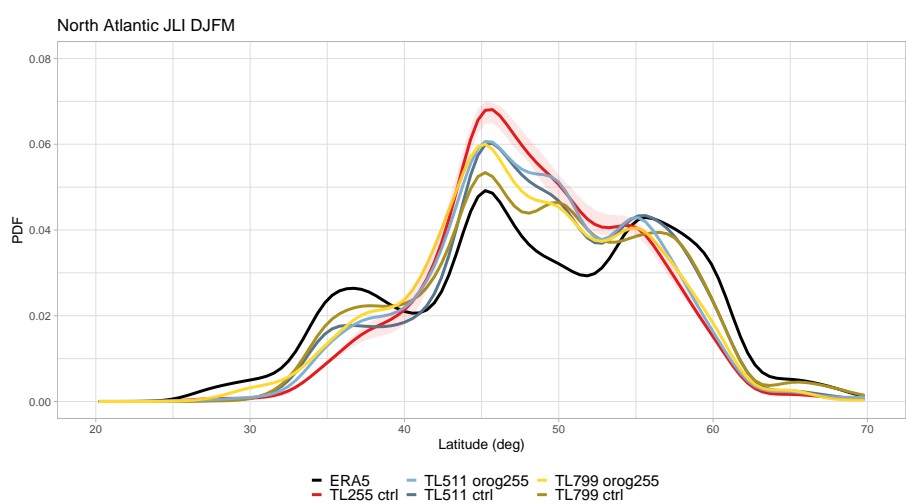

**Figure 3.** DJFM Jet Latitude Index distribution for the different EC-Earth3 experiments, for ERA5 (black), TL255 (red), TL511 (blue) and TL799 (yellow). Lighter colours indicate orog255 experiments. For TL255-ctrl run, the ribbons show the spread among the three integrations, with the ensemble mean in bold. Distributions are obtained from daily data by a kernel density estimation based on Gaussian smoothing with a bandwidth parameter of 1.25°.

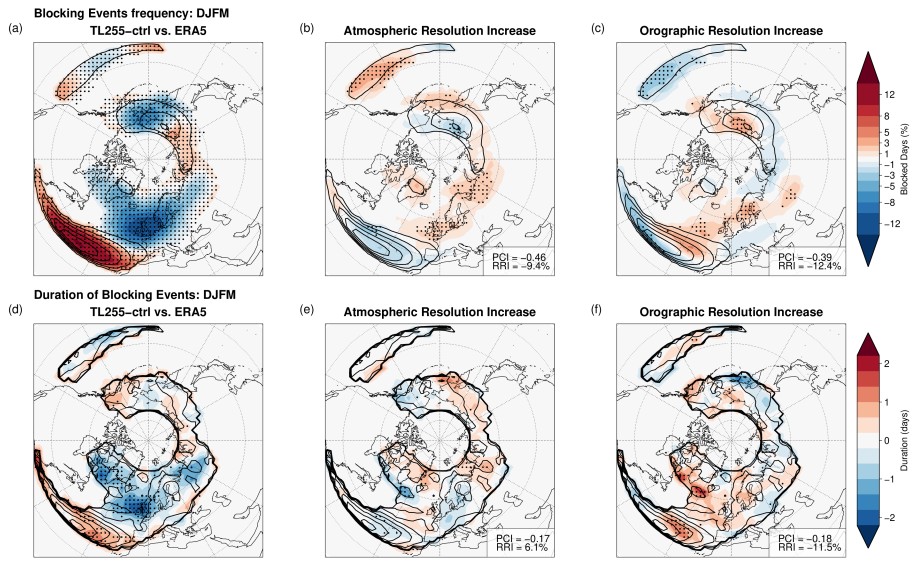

**Figure 4.** DJFM blocking events frequency (top row) and blocking events duration (bottom row) for (a,d) EC-Earth3 TL255-ctrl bias with respect to ERA5 (b,e) changes induced by the atmospheric resolution increase and (c,f) changes induced by the orographic resolution increase. Shading shows differences, contours the TL255-ctrl field. For Blocking Events frequency, please note the irregular color-bar spacing. For blocking frequency, contours are drawn every 5%. For blocking duration, contours are drawn every 0.5 days. Stippling indicates significance with a Welch t-test at the 5% level. In (b,c,e,f) the PCI and RRI (see text for details) are reported at the bottom left of each panel: large negative values implies reduced bias.

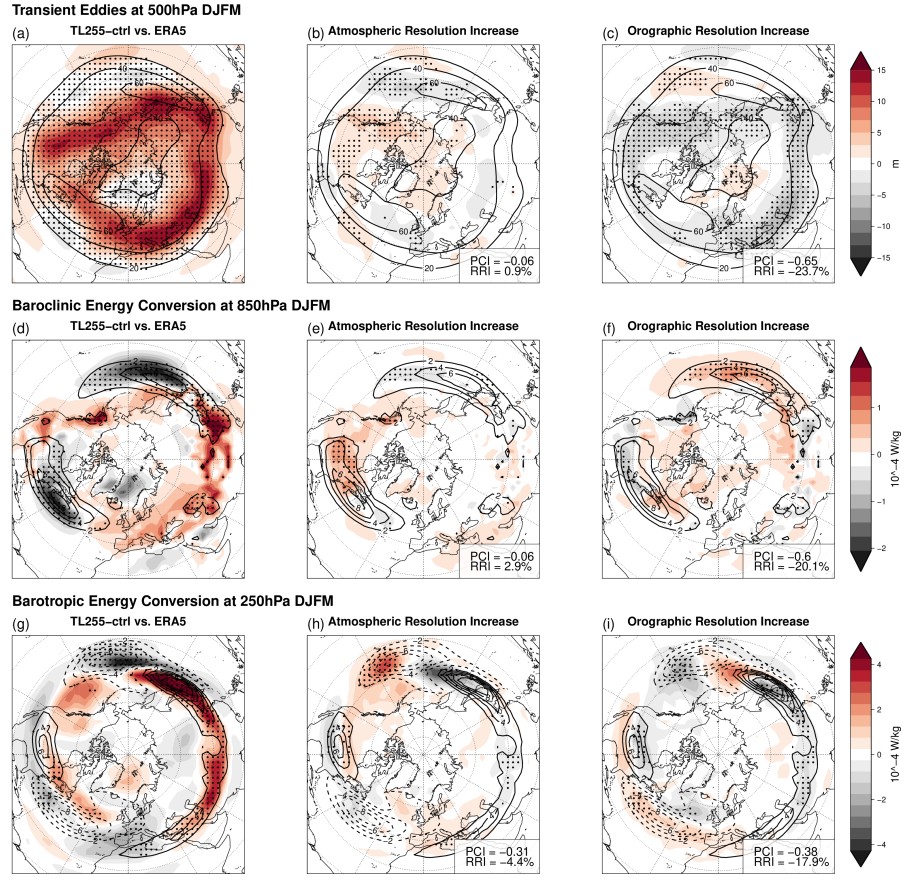

**Figure 5.** DJFM mean transient eddies standard deviation (top row), low tropospheric baroclinic energy conversion (middle row) and upper level barotropic energy conversion (bottom row) for (a,d,g) EC-Earth3 TL255-ctrl bias with respect to ERA5 (b,e,f) changes induced by the atmospheric resolution increase and (c,f,i) changes induced by the orographic resolution increase. Shading shows differences, contours the TL255-ctrl field. Stippling indicates significance with a Welch t-test at the 5% level. In (b,c,e,f,h,i) the PCI and RRI (see text for details) are reported at the bottom left of each panel: large negative values implies reduced bias.

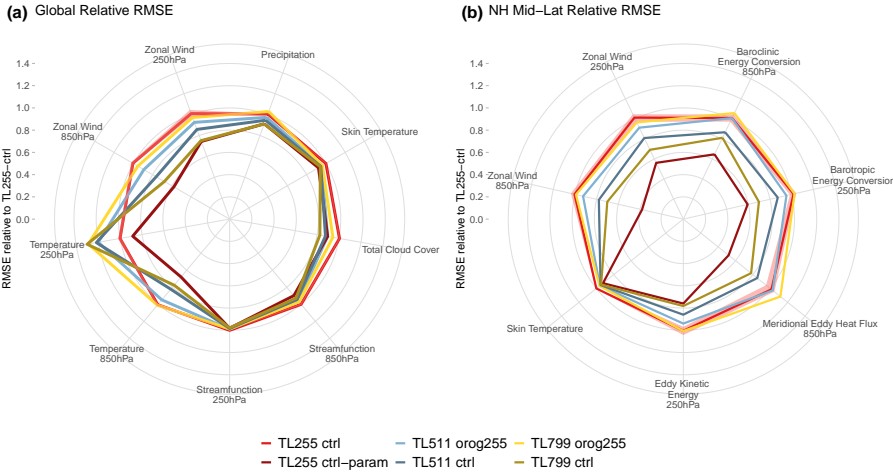

**Figure 6.** Radar chart showing the DJFM RMSE against ERA5 relative to the EC-Earth3 TL255-ctrl experiment for (a) a set of globally averaged fields and (b) a set of Northern Hemisphere mid-latitude (30°N-75°N) fields. Since the TL255-ctrl experiment is the reference (red), its values are always 1. TL511 is shown in blue and TL799 in yellow, with lighter colours indicating orog255 experiments. For TL255-ctrl run the three integrations available are shown, with the ensemble mean in bold. Values closer to the center of the plot imply smaller RMSE. Please note that (a) and (b) include different variables.

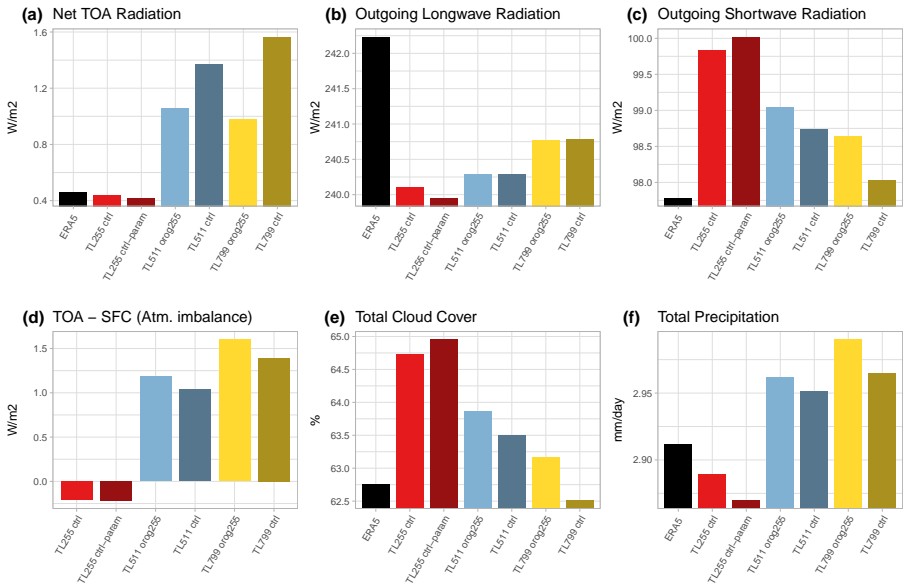

**Figure 7.** Global yearly-averaged mean (a) net TOA radiation (b) Outgoing Longwave radiation at TOA (c) Outgoing Shortwave radiation at TOA (d) Atmosphere imbalance (e) Total cloud cover (f) Total precipitation in the EC-Earth3 runs. TL511 is shown in blue and TL799 in yellow, with lighter colours indicating orog255 experiments

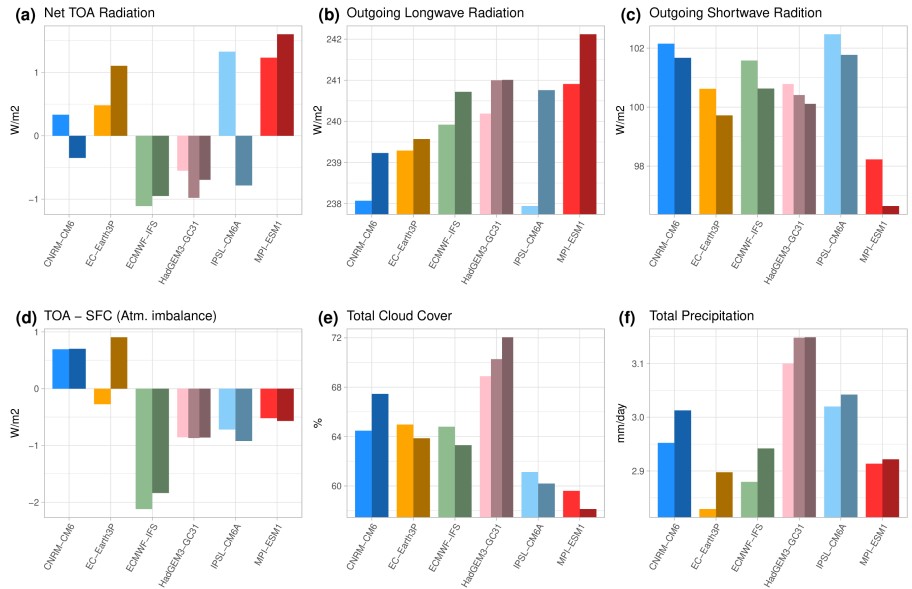

**Figure 8.** Global yearly-averaged mean (a) net TOA radiation (b) Outgoing Longwave radiation (c) Outgoing Shortwave radiation (d) Atmosphere imbalance (e) Total cloud cover (f) Total precipitation in HighResMIP atmosphere-only simulations. Darker colours indicate higher horizontal resolutions.