# Peer review of "Orographic resolution driving the improvements associated with horizontal resolution increase in the Northern Hemisphere winter mid-latitudes"

_Weather and Climate Dynamics, 2021_

## Referee Comment (RC2)

**Orographic resolution driving the improvements associated with horizontal resolution increase in the Northern Hemisphere winter mid-latitudes** by Paolo Davini, Federico Fabiano, and Irina Sandu

The study presents a systematic analysis of the effects of the small-scale orography on the large-scale circulation and in particular, the blocking phenomena. Due to coarse model resolution, climate models do not resolve the small-scale orographic features in sufficient detail. These features can impact the large-scale eddy forcing of the atmosphere with consequences for frequency and evolution of atmospheric blocking events resolved by these models. Using a set of carefully designed climate model integrations, the study demonstrates that increasing the orographic resolution in models, more so than increasing the atmospheric resolution, leads to a reduction of biases in model blocking frequency, jet latitude and eddy forcing (as compared to ERA5). Ultimately, the study concludes that properly tuning physical parameterizations in high resolution climate runs is key to eliminating potential energy imbalances caused due to changing resolution, and to fully reaping the benefits accompanying increased model resolution.

The experimental setup is scientifically sound, and the figures complement the discussion well. I do find the conclusions derived, at places, to be potentially misleading and slightly exaggerated, not being fully consistent with what the figures suggest. I address some of the concerns below. However, overall, the study could form an important contribution to the existing body of literature, and can be of value to HighResMIP and other climate model intercomparison projects. I recommend the manuscript for publication **pending major revisions**, after the following concerns have been addressed.

- 1. Impacts on Blocking: The T511 and T799 runs are integrated without the TOFD and SSO scheme and with other parameterizations tuned at a T255 resolution only. The parameterizations are not fully accurate (and not even accurately tuned in some instances) and in most cases respond to the climatology generated by the model. The climatology generated by the T511 and T799 are notably different (Figure 1 and 2). Given this, how do the authors isolate/remove the significant systematic errors that can be introduced by the untuned parameterizations? Moreover, it would be great if the authors can discuss how linear the response is between simply changing the atmospheric resolution and changing both the atmospheric resolution in concert.
- 2. Scaling: The changes due to changing resolution in Figure 4(b,c) and Figure 4(e,f) are much weaker when compared to the T255-ctrl biases in Figure 4(a) and Figure 4(d) respectively. If computationally possible, can the authors also illustrate the biases for the tuned T255-ctrl-param run as well (and if they are very different despite similar TOA budget)?

Moreover, given the current magnitude of the biases, some of the conclusions in Section 4 could be misleading. For instance, the changes due to increase in atmospheric resolution vs orogoraphic resolution appear comparable to me (and not strikingly different as claimed). Moreover, the magnitudes resulting from a 3-4 fold increase in horizontal resolution seem to be small, and even qualitatively, both types of resolution increases lead to a reduction in biases over Scandinavia and parts of Azores. I suggest the authors to please re-consider (mellow) such strong claims based on the experimental setup and, appropriately rewrite the findings.

Finally, can the authors provide some insight into how these biases are expected to scale with horizontal resolution? The planetary and synoptic scales are very well resolved at T255 resolution as well. Given this, is there a critical resolution, by which eddy feedback of the small-scale onto the large-scale, and its impact on blocking, is completely resolved? I appreciate that this could be an open question with no clear answer as of now, but given that the relative reduction in biases even after a 3x resolution increase is moderate at best, to what extent do the conclusions from this study confirm the impact of small-scales in influencing large-scale blocking?

3. Model Tuning: The final sentence of the abstract stresses that the findings point to the importance of tuning. This conclusions will be more readily acceptable if the authors could, if computationally possible, provide a comparison of their findings with a tuned version of T799. The parameterizations employed in the model are not scale aware and the improvements in the T511-ctrl and T799-ctrl run are not significant. Thus, to be able to confidently claim that model tuning is imperative to the blocking frequency, it would help to show the bias reduction in a tuned T799 run.

In addition, can the authors comment on how the parameterizations in the T255ctrl-param were tuned? For instance, what constraints were used, and was it tuned to produce realistic winds in the Northern hemisphere only or to optimize the TOA radiative budget?

4. **Mechanism:** Based on my understanding, a mechanism explaining how small-scale changes due to a more resolved orography project onto the planetary-scale does not clearly stand out from the study. The biases in transient forcing are in-phase with improvements in baroclinic energy conversion over the Pacific and North America, but not over the Tibetan Plateau. Also, the improvements in barotropic energy conversion over the Atlantic are more robust than actual impovements in blocking frequency.

Another concern I have is that the changes in eddy forcing with changes in resolution should be correlated with the TOA imbalance induced by resolution change (in cloud cover, precipitation etc). Moreover, the changes in the stationary wave forcing (as mentioned around L270) are similar when either the atmospheric resolution is increased or the orographic resolution in increased (Figure 2(b,c)). So, it is not totally clear to me how it is the changes around the orography that project onto the Atlantic jet stream, far away from the orography. I appreciate that it is not the focus of the study, but it would be great if the authors can provide some explanation for this for the general reader.

**Other Comments**

- 1. L15 : which 'has' resulted
- 2. L21 : Please provide an appropriate citation, if possible
- 3. L30 : Please elaborate more on the first sentence providing reasons as to why the effective resolution is lower. For example hyperdiffusion damping things grid-scale dynamics etc.
- 4. L43 : well-known
- 5. L52 : 'Thus, ' many questions related to ... remain open.
- 6. L58 : can remove the parenthesis to improve readability.
- 7. L60 : (this paragraph) suggests that previous studies have already established the role of a more resolved orography. Thus, by simply reading the paragraph, it is unclear how this study improves upon the findings of Kanehama et al. (2019). Please elaborate some key limitations of the previous study and how those are addressed in this new study providing more details than just 'in a more quantitative manner'.
- 8. L71 : Section 3 'and Section 4' include the analysis ... variability 'respectively'.
- 9. L81 : uppermost one  $\rightarrow$  model top
- 10. Section 2.1 : Please provide the vertical grid resolution in the troposphere.
- 11. L86 : is a one-year spinup sufficient? If I understand correctly, comprehensive model runs with a seasonal cycle need to be spun-up longer when focusing on climatological statistics.
- 12. L109 : Why is a three-member ensemble employed only for the T255 resolution integrations?
- 13. L117 : I am not sure if I understand "protocol" in this context. How about "project" perhaps.
- 14. L127 : Please remove 'are presented ... paragraphs.'
- 15. L131 : whose 'accurate' simulation still 'present a challenge' for ...

- 16. L140 : (paragraph) Please mention in some reasonable detail how sensitive the results are to the definition of blocking itself. The changes in blocking frequency over mainland Europe are not very significant can those changes be sensitive to the definition?
- 17. L151 : investigating
- 18. L171 : Similar to the
- 19. L179 : lower middle troposphere? please be more specific by providing the rough pressure level you are referring to (500 hPa? 750 hPa?)
- 20. L177 : reference pressure
- 21. L153 : standard dev wrt the zonal mean? Please be more specific.
- 22. From a reabability point of view, I think it would be better to swap section 2.2 and 2.3 and place the (current) section 2.3 right after the model setup. This will ensure all the model runs specific details are discussed together.
- 23. L206 : There is no noticeable shift in the jet height. Please add the old and new tropopause height to support the claim.
- 24. L210 : larger 'orographic' wave activity, I assume?
- 25. L215 : How is the streamfunction defined? Is it simply the geopotential?
- 26. L245 : Please mention the key reasons why models have such a modality bias.
- 27. L251 : While I agree that in a quantitative sense T799 is closest to ERA5, it also develops a fourth peak around 50°N and has the northmost peak around 57°N as opposed to ERA5 and T511 which has similar structure in terms of modality and gradients. Can you please comment more on this, why the distribution does not converge qualitatively with increases in atmospheric resolution?
- 28. L257 : please remove 'obviously'
- 29. L261 : I do not fully agree with the conclusion. Especially over mainland Europe, increase in orography resolution worsens the bias. In addition, over the UK and Scandinavian regions, changes due to atm res. are comparable (and in cases larger) than the changes due to oro. res. Therefore, the statement is inarguably true only for the lower latitude Azores Atlantic basin.
- 30. L268 : Again, I do not fully agree with the conclusion. The changes in blocking duration are definitely comparable to the bias between T255 and ERA5, along with worsening of the bias over mainland Europe. While I support the conclusion over the Antantic, concluding the same for mainland Europe is not so stratightforward.

- 31. L270: The stationary patterns by changes in atmos. res and orog. res. are affected similarly (Figure 2b,c). Thus I think the final statement of the paragraph should be framed more carefully.
- 32. L275 : Is the bias significantly stronger than for the tuned T255 run?
- 33. Figure 5 : Is the large red band over Eurasia in the bias (Figure 5a) explained exclusively by the Alps? (similar to the bias over western North America)
- 34. L283 : ... orographic resolution. The changes to the ... I suggest a paragraph break here.
- 35. L286 : same terms (not shown)?
- 36. L295 : This is not totally clear to me, can you please explain why?
- 37. L317 : significantly for almost
- 38. L357 : I think it decreases by 1% at T511 and by 2% at T799 instead. Isn't it?
- 39. L396 : ... might be considered responsible for the absence ...
- 40. Figure 1(b) : please mention the maximum temperature difference associated with stratospheric cooling. The colorbar saturates early. It would also be great if the authors can mention in the Figure 4's caption the maximum value of the biases in Figure 4a as the colorbar saturates early for lower values. More generally, please increase the font size in all the figures, and wherever the colorbar saturates, provide the maximum value of the statistic.
- 41. How does the Jet Latitude Index (JLI) compare for the different models integrations over the Pacific? This is part of my broader concern, regarding how the wind changes due to flow blocking around the Rocky Mountains and Tibetan Plateau relate to changes in the Atlantic JLI.

---

## Author Comment (AC1)

Dear Editor and dear Reviewers,

Thank you for your comments and observations, which helped us to considerably improve the quality of our manuscript. We made several minor adjustment and a few major changes, which are listed here below:

- We evaluated the statistical significance with a Welch t-test in Figure 1,2,4,5 making use of the assumption of independent winters.
- We introduced two scalar measures to evaluate the improvement or the deterioration following the orographic and atmospheric resolution changes: the pattern correlation improvement and the relative RMSE improvement. These are now described in Section 2.4 and used in Figure 1,2,4,5.
- We improved the introduction section in order to stress out the relevance of tuning activities in the Earth System Modeling development, as well as the sensitivity of the hydrological cycle, cloud dynamics and of the radiative budget following atmospheric resolution changes.
- We added a supplementary material section introducing four more figures which aims at clarifying some aspects of blocking analysis (introducing a second blocking index based on the anomaly from the mean flow) and of the sensitivity to atmospheric/orographic resolution increase.

We hope that with these changes you will find the paper now suitable for publication in the Weather and Climate Dynamics.
You may find a detailed reply to the reviewers' comments in the following pages.

Best regards,

Paolo Davini

**REVIEWER 1**

**This paper studies and contrasts impacts of increasing atmospheric model resolution vs increasing the resolution of orography on the NH wintertime mid-latitude atmospheric dynamics in a general circulation model. As climate modelling moves towards higher and higher resolution, studies like this are important and interesting to the community. The simulations are appropriate and the analysis is clear and well presented, with key implications for future research. I am keen to see this paper published in Weather and Climate Dynamics. I have a few more significant comments that should be addressed prior to publication, followed by some minor suggestions.**

**MAJOR COMMENTS**

**I think you need to justify more clearly, and discuss in more detail the implications of, your choice to turn off the subgrid scale parameterizations, since this is not representative of resolution changes in climate models.**

We would like to thank the reviewer for this comment since it helped us to improve the discussion about this important experimental choice. In the previous version of the manuscript it was not clear enough that TOFD and SSO parametrizations are actively interacting between each other, and they are influenced by the mean orography since both are dependent on the strength of the mean wind. Furthermore, the two schemes massively rely on subgrid orography which - although is derived from the same orography dataset - is resolution dependent as well. Although it is numerically possible to turn the orographic parametrizations on and change the mean orography resolution, this would produce a series of spurious interaction effects due to the sensitivity of the mutual interaction of parametrizations with resolution, so that it would be hard to disentangle the different contributions.

In the new version of the manuscript (Lines 113-117) we better explain why this passage is mandatory - as done by Kanehama et al. 2019.

**Part of the discussion of implications should include the fact that this study almost certainly overestimates the importance of orographic resolution because by turning off the parameterizations:**
**- Zonal winds overestimated and therefore role of orography overestimated**
**- Parameterizations are designed to fill the unresolved gap, so they should be, by definition, less important at higher resolutions. The true importance of orographic resolution in climate models lies somewhere between your results, and the case if the parameterizations were perfect (which they are not!), which is that there should be no difference between high and low resolution orography.**

Although it is true that the zonal winds are overestimated in our runs, we partially disagree with the reviewer on the point concerning the overestimation of the role of mean resolved orography. Assuming (in theory) that no other biases are operating on the mean wind, the positive wind speed bias we see in our experiments is due to the underestimation of the orographic drag. This makes sense considering that the orographic resolution is too coarse to represent the entire spectrum of orographic-induced wave perturbations. The goal of the orographic parametrizations is exactly to fill the gap between the mean orography at coarse

resolution and the actual orography, so that we expect them to be more important at lower resolution. However, in the current work we are actually studying the impact of orographic resolution (i.e. the role of the resolution of the mean resolved orography, as detailed in the text) not the role of orography per se.

This latter point is of course - as stated by the reviewer - much more complex and should also include the effect of the parametrized orographic drag. This is indeed one of the reasons why we turn off the orographic parametrizations, in order to avoid mixed effects which will affect any conclusion. In this way, the impact on the flow of the resolved orography does not depend on the presence/absence of TOFD and SSO schemes.

**If feasible I also highly recommend, as in Kanehama et al. (2019), that you repeat one of your high resolution experiments with the subgrid scale parameterizations on, to confirm that your main conclusions hold true even when the parameterizations are included (though for the above reasons I would expect the improvement with increased orographic resolution to be reduced).**

Although we agree with the reviewer and we really would like to have such integration as soon as we can, a single run at TL799 costs about 600k core hours, about 30 times more than an equivalent integration at TL255. Unfortunately we do not have the computational resources to run a high resolution configuration at TL511 and TL799 with the orographic parametrizations active at the moment. We are planning to perform such integration in an upcoming numerical project.

**I think the last section could be structured more clearly to make your conclusion clearer and more concise. I agree with your conclusion that re-tuning high resolution models is important, but I have a couple of concerns:**
- **I'm not convinced you have shown the link to the biases in atmospheric dynamics sufficiently clearly – this requires more analysis, or perhaps emphasis on the differences between the TL511_orog255 and TL799_orog255 in figure 6 (including analysis to confirm that the differences are statistically significant, see comment 3), to illustrate the impact of the radiative budget biases on the dynamics, which is the focus on this paper (as stated on line 68).**

In the new version of the manuscript we further stressed out the differences between TL511-orog255 and TL799-orog255, and a comparison of some of the most important fields are now shown in the supplementary material Figure SM3. This clearly shows, making use also of the new scalar diagnostics PRI and RRI, that the TL799-orog255 deteriorates the mean climate if compared to the TL511-orog255. Moreover, we highlighted that radiation is not "the driver" of the changes but rather a proxy of the changes driven by cloud microphysics and convection parametrizations. Of course, we are aware that a more comprehensive analysis, perhaps including extra experiments for a tuned configuration or with more resolution alternatives, could provide more robust conclusions.

Overall, we believe that the connection between the precipitation changes, radiative fluxes and the resulting pattern of temperature and wind is quite clear from Figure 6. The new Figure 6 now also includes a measure of the error using the three ensemble members of the T255-ctrl experiments, so that it shows the robustness of the changes discussed.

- **You have shown the radiative budget impact clearly, but your introduction is missing a lot of the literature on this if the radiative budget is to be your focus (e.g. https://link.springer.com/article/10.1007/s00382-018-4547-y, https://agupubs.onlinelibrary.wiley.com/doi/10.1029/2019JD032184, https://www.jstage.jst.go.jp/article/jmsj/98/1/98_2020-005/_article, https://journals.ametsoc.org/view/journals/bams/98/3/bams-d-15-00135.1.xml to list a few recent papers). Alternatively (the option I would recommend) you could limit your analysis more to the dynamics, spend less time discussing the exact differences in the radiative budget, and in one or two paragraphs summarize these results and the importance of tuning (including reference to previous papers that make this argument too), particularly for the dynamics (the point you are making).**

We agree with the reviewer that the introduction on the impact on radiative budget and hydrological cycle was too sketched. In the new version of the manuscript we expanded the introduction (L36-49) and the section which discusses the radiative budget making use of several references - among which the ones suggested - to better motivate our discussion on the radiation budget. The connection between precipitation, cloud cover and dynamics and radiative budget is also clearly highlighted.

- **After a strong focus on dynamics in the rest of the paper I was a little surprised by the extensive analysis of the radiative budget in section 6 until I read the discussion section. I agree with your assessment that this is likely related to the lack of tuning for the higher resolutions. I recommend motivating the analysis section a bit more with the discussion about typical high-resolution initiatives lacking tuning for the high-res version that you discuss in lines 395-396 – either at the beginning of section 6, or in the introduction.**

As above, we agree with the reviewer that the discussion of tuning - given its relevance for the paper - was way too approximative. In the new version we expanded the discussion (L44-49) on the tuning moving it in the introductory section in order to make clear that this is one of the key points that has to be discussed when increasing atmospheric resolution.

**Statistical significance - some of your changes are quite small, e.g. the changes in zonal wind in Fig. 2, and differences between TL511_orog255 and TL799_orog255 in figure 6 – could you give an estimate of the fractional change, and/or statistical significance of these changes, perhaps based on assuming independence between consecutive winters (A reasonable assumption for climatological fixed SSTs)?**

We implemented a Welch t-test for most of the figures exploiting the independence between consecutive winters, as suggested. This is now represented by stippling in Figure 1,2,4,5. For mean climate changes (Figure 1 and 2) significance levels are at 99%, while for variability it is shown at 95% (Figure 4 and 5). Regarding Figure 6, this kind of approach might limit the readability of the plot itself, so to provide an estimate of the significance of the results we preferred to implement the same methodology of Figure 3, i.e. showing the variability of the 3 ensemble members available for the TL255 ctrl integrations. All the results point to very robust findings in terms of differences but for atmospheric blocking, so that we toned down our statement in that section.

Furthermore, as mentioned above, we introduce two scalars diagnostics (PCI and RRI) which provide a synthetic measure of the mode improvement.

**Minor comments**

**Line 72: Sections 3 and 4 present analysis of the mean climate and of the mid-latitude variability respectively.**

Thanks, this has been corrected.

**Line 135. Define GHGS and GHGN to make it easier for readers (e.g. geopotential height gradient at a southern (GHGS) and northern (GHGN) latitude)**

Thanks, this has been corrected.

**Line 227. You have never explicitly mentioned that the increase in resolution will result in an increase in maximum heights of mountains – this would be good to add (https://doi.org/10.1029/2020AV000343 might be an interesting paper to reference)**

This is correct, the relevance of orography maximum height is now mentioned in the introduction.

**Lines 245-255. The improvement of the tri-modal distribution with increased orographic resolution is perhaps consistent with this paper: https://agupubs.onlinelibrary.wiley.com/doi/abs/10.1029/2019GL084780**

Even though certainly the Greenland mass is fundamental for shaping North Atlantic variability, we are not sure this is actually the case - at least in the EC-Earth climate model - since Greenland orography is only partially affected by the changes in resolution given its "smoothness", as can be seen from the Figure X1 here below (see the dome at about 75N). At TL255 the mean resolved orography over Greeland is already decently defined. While increasing the orographic resolution increases the maximum orography over the North America continent by hundreds of meters in the region of the Mexican Plateau and of the Rocky Mountains, no evident changes are seen over Greenland. More specific sensitivity tests should be performed to provide some more conclusive results on which region of the globe is more relevant for shaping the jet variability, but this goes beyond the goal of the current study.

[Figure]

*Figure X1: Maximum orography over Central and North America.*

**Figure 3. I'd be interested to see your TL255_ctrl-param simulation added to this.**
As shown in Figure X2 here below, the TL255-ctrl param shows a good representation of the trimodal North Atlantic, which is comparable to the high resolution configuration. This is expected, since the orographic parametrization simulates the missing effect of smaller orographic scales.

[Figure]

*Figure X2: As Figure 3, but including also TL255-ctrl-param*

Although we understand the importance of showing also the TL255 run including the parametrization, we would prefer to avoid this plot in the main manuscript since we would like to avoid distracting the reader with the comparison between the orographic parametrizations and the other integrations.

**Line 255: Blocking and the tri-modal distribution of the Atlantic jet are not necessarily separate modes of variability (e.g. https://rmets.onlinelibrary.wiley.com/doi/10.1002/qj.959, https://rmets.onlinelibrary.wiley.com/doi/10.1002/qj.3155).**
We agree with the reviewer, blocking activity is partially associated with the Atlantic jet variability: especially blocking activity over Greenland is associated with cyclonic wave breaking and negative NAO. However, the coupling with jet variability is rather weak if the focus is moved on European blocking frequency, since Central European/Scandinavian blocking is usually detached from jet oscillations. We clarified that the two things are not completely independent now in the new version of the manuscript.

**Line 255. I'm curious how much of this bias is because of turning off the sub-grid scale orography parameterization.**
This is now included in Supplementary Material Figure SM1, together with the evaluation of atmospheric blocking making use of anomaly-based index as done by Schwierz et al 2004 (Figure SM2). Panel b and f show the improvement induced by the orographic parametrization for blocking events and duration.

**Line 270. Why only stationary waves and not also the role of the zonal wind changes?**
This sentence has been rephrased.

**Line 290. I don't see this equatorward bias – the negative anomalies seem to line up with the negative values in the ctrl experiment and the positive anomalies with the positive value in the ctrl, which suggests a magnitude bias rather than a latitudinal shift. Perhaps this latitudinal shift is present towards the end of the Atlantic jet stream, as suggested by your following comment on the tilt of the Atlantic jet, but that is not the dominant pattern in this figure. Please clarify which pattern/location you are referring to.**
We thank the reviewer for pointing this issue out, since we did not specify that we were discussing the Atlantic sector and not the Pacific one. This has been addressed in the new version of the manuscript.

**Figure 5. Mention what the black contours are.**
Thanks, this has been corrected.

**Figures 3 and 6. It took me a while to notice that there were colour groups for the high and lower resolution topography – this is useful, and I would recommend informing the reader you have done this, and perhaps making it clearer by using different line styles to differentiate between high and low resolution topography versions. This would emphasise the benefit of increasing orographic resolution. Also, please consider colour-blind readers: asking readers to distinguish between red and green is not the best for those readers (e.g. https://www.ascb.org/science-news/how-to-make-scientific-figures-accessible-to-read ers-with-color-blindness/)**
Thanks a lot for the suggestion. We updated the color scheme and mentioned now in the caption that different color groups refer to different resolution sets.

**Line 371. As you have pointed out the lack of tuning for the higher resolution experiments, which will particularly affect precipitation and cloud cover, I think it is hard to make robust conclusions about the different basins.**
We definitely agree with the reviewer that this point is a bit too speculative, therefore we decide to remove this paragraph.

**Line 377: Suggested re-wording: "the deterioration of the radiative budget counteracts any potential improvements provided by the refinement of the atmospheric grid" – the original wording of "most of" implies that there is a definite improvement in the dynamics with the increased atmospheric resolution, which I don't think you have shown as you don't have a re-tuned simulation.**
Thanks a lot, this is a clearer alternative to our original wording.

**Line 397 – Suggested re-wording: potential mechanism responsible …. is increased in our experiments.**
We updated the text as suggested.

**REVIEWER 2**

**The study presents a systematic analysis of the effects of the small-scale orography on the large-scale circulation and in particular, the blocking phenomena. Due to coarse model resolution, climate models do not resolve the small-scale orographic features in sufficient detail. These features can impact the large-scale eddy forcing of the atmosphere with consequences for frequency and evolution of atmospheric blocking events resolved by these models. Using a set of carefully designed climate model integrations, the study demonstrates that increasing the orographic resolution in models, more so than increasing the atmospheric resolution, leads to a reduction of biases in model blocking frequency, jet latitude and eddy forcing (as compared to ERA5). Ultimately, the study concludes that properly tuning physical parameterizations in high resolution climate runs is key to eliminating potential energy imbalances caused due to changing resolution, and to fully reaping the benefits accompanying increased model resolution. The experimental setup is scientifically sound, and the figures complement the discussion well. I do find the conclusions derived, at places, to be potentially misleading and slightly exaggerated, not being fully consistent with what the figures suggest. I address some of**
**the concerns below. However, overall, the study could form an important contribution to the existing body of literature, and can be of value to HighResMIP and other climate model intercomparison projects. I recommend the manuscript for publication pending major revisions, after the following concerns have been addressed.**

**MAJOR COMMENTS**

**1. Impacts on Blocking: The T511 and T799 runs are integrated without the TOFD and SSO scheme and with other parameterizations tuned at a T255 resolution only. The parameterizations are not fully accurate (and not even accurately tuned in some instances) and in most cases respond to the climatology generated by the model. The climatology generated by the T511 and T799 are notably different (Figure 1 and 2). Given this, how do the authors isolate/remove the significant systematic errors that can be introduced by the untuned parameterizations?**
We definitely agree with the reviewer on this point. Indeed, we cannot disentangle the impact of the untuned parametrizations - on the numerical implication induced by increasing the horizontal resolution - from the benefits that high resolution can bring. This is actually our main conclusion, since we highlight that the untuned model offsets a large part of the benefits by introducing biases in the precipitation, cloud dynamics and cover which impact the radiative budget. We agree with the reviewer that tuning high resolution models is fundamental to assess the real benefit of increased resolution.

**Moreover, it would be great if the authors can discuss how linear the response is between simply changing the atmospheric resolution and changing both the atmospheric resolution and orographic resolution in concert.**
Regarding linearity, the assumption we made in our work is that the resolution and orographic signals are linear so that we can operate our "decomposition" analysis. This has now been specified in the text.

**2. Scaling: The changes due to changing resolution in Figure 4(b,c) and Figure 4(e,f) are much weaker when compared to the T255-ctrl biases in Figure 4(a) and Figure 4(d) respectively.If computationally possible, can the authors also illustrate the biases for the tuned T255-ctrl-param run as well (and if they are very different despite similar TOA budget)?**

This integration is available, and it has been reported in Figure 6. The massive improvement induced by the TOFD and SSO schemes is now shown in Figure SM1. The magnitude of this improvement is larger than the ones observed for orographic and resolution alone. This also highlights how the orographic parameterizations are relevant for blocking representation.

**Moreover, given the current magnitude of the biases, some of the conclusions in Section 4 could be misleading. For instance, the changes due to increase in atmospheric resolution vs orographic resolution appear comparable to me (and not strikingly different as claimed). Moreover, the magnitudes resulting from a 3-4 fold increase in horizontal resolution seem to be small, and even qualitatively, both types of resolution increases lead to a reduction in biases over Scandinavia and parts of Azores. I suggest the authors to please re-consider (mellow) such strong claims based on the experimental setup and, appropriately, rewrite the findings.**

Following the reviewer's comments, we adjusted our discussion of impact on blocking induced by the orography and resolution, since the differences are not so striking as we initially reported. The newly introduced PCI and RRI diagnostics helps us in quantifying the changes, showing that while changes for the blocking frequency are comparable, when looking at blocking duration the improvements introduced by orographic resolution are slightly  larger. It must be anyhow underlined that the statistical significance is limited due to the large interannual  variability of blocking. We also introduced a second blocking index, based on the anomaly from the mean flow, which shows larger benefit from the orographic increase than from the resolution increase and that - as shown in Figure SM2 - highlights that the blocking signal seems to be sensible to the blocking index adopte. All these points are now mentioned in the text.

**Finally, can the authors provide some insight into how these biases are expected to scale with horizontal resolution? The planetary and synoptic scales are very well resolved at T255 resolution as well. Given this, is there a critical resolution, by which eddy feedback of the small-scale onto the large-scale, and its impact on blocking, is completely resolved? I appreciate that this could be an open question with no clear answer as of now, but given that the relative reduction in biases even after a 3x resolution increase is moderate at best, to what extent do the conclusions from this study confirm the impact of small-scales in influencing large-scale blocking?**

We agree with the reviewer that this is a challenging and interesting issue. We cannot really say something about scaling: we see TL799 doing better than TL511 and that this is associated with orographic improvements, and some of the improvements are likely offset by the lack of tuning. We agree with the reviewer that the amount of the improvement is limited, as shown by RRI, with a bias reduction of only 10%, while activating the TOFD and SSO scheme brings about a 30% improvement.

Our speculation on this point is that the numerical and parametrization-induced issues that have been shown having an impact on precipitation/cloud dynamics and consequently on

the radiative budget are altering significantly the tropical processes so that a larger part of the improvement in blocking and other mid-lat features are negatively impacted. Rossby wave propagation from the tropics and properties of the Rossby wave guide are likely affected by changes induced by atmospheric resolution increase. Blocking is a largely non-linear process, so that those tropical changes can have a relevant impact. In this sense, this potentially explains why the parametrization has a strong impact on blocking, much larger than what found by orography and resolution, since TOFD and SSO operate directly on the mid-latitudes without altering any tropical aspect. However, such a strong claim can be only verified with a high resolution integration which is far beyond our possibilities so that we prefer to avoid this discussion in the text.

**3. Model Tuning: The final sentence of the abstract stresses that the findings point to the importance of tuning. These conclusions will be more readily acceptable if the authors could, if computationally possible, provide a comparison of their findings with a tuned version of T799.**
Although we agree with the reviewer this is certainly an interesting step to be carried out, and that we hope to be able to perform this analysis in a future project, unfortunately this is far beyond our computational possibilities. A single run at TL799 costs about 600k core hours, about 30 times more than an equivalent integration at TL255. On top of that, a proper tuning of the TL799 will require an enormous effort including a sensitivity analysis of the different tuning knobs, so that likely much more than 50 years of integration will be required.

**The parameterizations employed in the model are not scale aware and the improvements in the T511-ctrl and T799-ctrl run are not significant.**
Actually almost all the parametrization developed by ECMWF are scale-aware, keeping inside their code some features which depend on the horizontal resolution, the timestep or some external condition (as for sub-grid orography). Moreover, most of the improvements seen in our experiments are significant, as now shown in almost all the figures which report a Welch t-test at 95% or even 99% level. There are examples - such as the stratospheric cooling error which increases with horizontal spacing - which are not induced by the parametrizations rather by limitations in the numerics (Politchouk et al. 2019).

**Thus, to be able to confidently claim that model tuning is imperative to the blocking frequency, it would help to show the bias reduction in a tuned T799 run.**
As mentioned above, this is not feasible with the current computational availability. However, we mentioned more clearly this limitation in the discussion part.

**In addition, can the authors comment on how the parameterizations in the T255-ctrl-param were tuned? For instance, what constraints were used, and was it tuned to produce realistic winds in the Northern hemisphere only or to optimize the TOA radiative budget?**
EC-Earth3 has been tuned aiming at having a proper surface radiative budget and surface temperature. More details can be found in Doescher et al. 2021 (https://doi.org/10.5194/gmd-2020-446).

**4. Mechanism: Based on my understanding, a mechanism explaining how small-scale changes due to a more resolved orography project onto the planetary-scale does not clearly stand out from the study. The biases in transient forcing are in-phase with**

**improvements in baroclinic energy conversion over the Pacific and North America, but not over the Tibetan Plateau. Also, the improvements in barotropic energy conversion over the Atlantic are more robust than actual improvements in blocking frequency.**

Indeed, the impact of the mean orography on mid-latitude eddies is very complex and relies on complex non-linear eddy-mean flow interaction. The hypothesis we have is that mean climate changes - as having a weaker jet - will favor an increase in the frequency of RWB events which modifies the structure of the mean flow. The very different orography-jet interaction between Tibetan Plateau and Rocky mountains is thus not surprising, given the different shape and orientation of the two mountain barriers.

**Another concern I have is that the changes in eddy forcing with changes in resolution should be correlated with the TOA imbalance induced by resolution change (in cloud cover, precipitation etc). Moreover, the changes in the stationary wave forcing (as mentioned around L270) are similar when either the atmospheric resolution is increased or the orographic resolution is increased (Figure 2(b,c)). So, it is not totally clear to me how it is the changes around the orography that project onto the Atlantic jet stream, far away from the orography. I appreciate that it is not the focus of the study, but it would be great if the authors can provide some explanation for this for the general reader.**

The dynamics of the North Atlantic is a result of the eddy-mean flow interactions, so that as we have shown in Figure 5 the eddies dynamics is considerably affected by the changes in the orography. Our interpretation is that although the upper stationary wave pattern is influenced in a quite similar way either by the atmospheric and orographic resolution (while in the lower troposphere the orographic impact is larger), the orographic drag has a relevant impact on the generation of baroclinic eddies. These propagate along the jet stream leading to a weaker North Atlantic jet, which in turns favor more wave breaking. We noticed a systematic improvement when increasing orographic resolution (TL799 is better than TL511 than TL255) both in terms of RRI and PCI, while this is not seen for the atmospheric grid changes (as now shown in Figure SM3 and SM4). Although a detailed correlation analysis between the biases is something we tried to perform, with only 3 points it is impossible to draw any robust conclusions. It will be definitely intriguing to perform such an analysis making use of HighResMIP integrations, but this is beyond the goal of the current study.

**Minor comments**

**1. L15 : which 'has' resulted**

This has been corrected.

**2. L21 : Please provide an appropriate citation, if possible**

Many modeling initiatives have been focusing on increasing atmospheric resolution, and they are listed in the following sentence. However, for completeness we referred to the HighResMip project here.

**3. L30 : Please elaborate more on the first sentence - providing reasons - as to why the effective resolution is lower. For example - hyperdiffusion damping things grid-scale dynamics etc.**

The sentence has been rephrased to include a few reasons leading to a coarser effective resolution than the grid spacing, such as numerical diffusion and aliasing.

**4. L43 : well-known**

Corrected.

**5. L52 : 'Thus, ' many questions related to ... remain open.**

Corrected.

**6. L58 : can remove the parenthesis to improve readability.**

Corrected.

**7. L60 : (this paragraph) suggests that previous studies have already established the role of a more resolved orography. Thus, by simply reading the paragraph, it is unclear how this study improves upon the findings of Kanehama et al. (2019). Please elaborate some key limitations of the previous study and how those are addressed in this new study providing more details than just 'in a more quantitative manner'.**

The main difference is that in the current work we operate on climate timescales, so that we can provide a much deeper understanding of the relative impact of resolution and orography on a different set of phenomena, such as high frequency variability. This is now discussed in the final part of the introduction.

**8. L71 : Section 3 'and Section 4' include the analysis ... variability 'respectively'.**

Corrected, thanks.

**9. L81 : uppermost one → model top**

Thanks, this has been corrected

**10. Section 2.1 : Please provide the vertical grid resolution in the troposphere.**

We would prefer to avoid this discussion, since there is no unique answer to this comment: the model works with hybrid coordinates, meaning that in its upper vertical domain the model is using pressure level coordinates, as can be seen here: https://www.ecmwf.int/en/forecasts/documentation-and-support/91-model-levels
This implies that at the Equator, assuming a tropopause at 100hPa height, about 50 levels are in the troposphere, and they decrease to 35-40 at the Pole.

**11. L86 : is a one-year spinup sufficient? If I understand correctly, comprehensive model runs with a seasonal cycle need to be spun-up longer when focusing on climatological statistics.**

Considering that these are atmospheric only runs with fixed land use and vegetation we assume that 1 year is enough: the spin up time of the atmosphere is negligible (less than 1 month) and the drift induced by radiative changes is observed to occur in the first 2-3 months of integration. The only feature which might need a larger timescale is snow cover, but no evident trend has been observed in any of those runs.

**12. L109 : Why is a three-member ensemble employed only for the T255 resolution integrations?**
This is mainly due to computational reasons. As mentioned above, there is a massive difference between the requirements of a TL255 and TL511/TL799, so we used the low resolution configuration also to test the interannual variability.

**13. L117 : I am not sure if I understand "protocol" in this context. How about "project" perhaps.**
We agree with the reviewer and we corrected making use of project

**14. L127 : Please remove 'are presented ... paragraphs.'**
We would prefer to keep this introduction to present the following paragraphs

**15. L131 : whose 'accurate' simulation still 'present a challenge' for …**
Thanks, we implemented the suggested correction

**16. L140 : (paragraph) Please mention in some reasonable detail how sensitive the results are to the definition of blocking itself. The changes in blocking frequency over mainland Europe are not very significant - can those changes be sensitive to the definition?**
We would like to thank the reviewer for this suggestion. We computed the blocking analysis also with an anomaly based index, which is very similar to the one used by Woollings et al 2018 and is derived from the original Schwierz et al 2004 definition. This is now included in the Supplementary Material in Figure SM2. Although the results are barely significant over the European continent, the signal induced by orographic changes is even larger than for the reversal based index. This has now been mentioned in the text, together with a discussion on the differences between blocking indices and the need of a very long integration to highlight differences in those fields. Moreover, this might suggest that the impact of the orography on variability is larger than on the mean state.

**17. L151 : investigating**
Corrected.

**18. L171 : Similar to the**
The sentence has been rephrased.

**19. L179 : lower middle troposphere? please be more specific by providing the rough pressure level you are referring to (500 hPa? 750 hPa?)**
We make an explicit reference to 850hPa which is the level at which we evaluate the baroclinic conversion term.

**20. L177 : reference pressure**
Thanks for pointing this out, this has been corrected.

**21. L153 : standard dev wrt the zonal mean? Please be more specific.**
We are not sure we understand the reviewer's comment. Transient eddies at each grid point are computed as the standard deviation (in time) of the highpass filtered geopotential height

field, which is evaluated on a daily basis. The resulting field is the one shown in the top row of Figure 5. No zonal averaging is introduced. If the reviewer had something more specific in mind, we would be happy to improve our definition.

**22. From a readability point of view, I think it would be better to swap section 2.2 and 2.3 and place the (current) section 2.3 right after the model setup. This will ensure all the model runs specific details are discussed together.**
This is a brilliant suggestion, we swapped the two sections.

**23. L206 : There is no noticeable shift in the jet height. Please add the old and new tropopause height to support the claim.**
This is now shown in Figure 1, upper row. Although the resolution is not very fine - we are working on the standard 19 pressure levels from default CMIP6 - the tropopause increase is quite evident.

**24. L210 : larger 'orographic' wave activity, I assume?**
We added "orographic" to specify that this is induced by orography

**25. L215 : How is the streamfunction defined? Is it simply the geopotential?**
No, this is the standard streamfunction computed from the integration of the u and v field, as below:

$$u = -\frac{\partial \psi}{\partial y} \text{ and } v = \frac{\partial \psi}{\partial x}.$$

**26. L245 : Please mention the key reasons why models have such a modality bias.**
As now added in the text, models tend to have a too strong and poleward displaced jet which reduces the variability from a wobbling mode to a pulsing one (Barnes and Polvani 2013). This has also consequences on the way the North Atlantic Oscillation is represented.

**27. L251 : While I agree that in a quantitative sense T799 is closest to ERA5, it also develops a fourth peak around 50∘N and has the northmost peak around 57∘N as opposed to ERA5 and T511 which has similar structure in terms of modality and gradients. Can you please comment more on this, why the distribution does not converge qualitatively with increases in atmospheric resolution?**
First of all, we need to take into consideration that PDFs as the ones shown in Figure 3 are certainly affected by the bandwidth of the kernel density estimator, so that the presence/absence of a fourth peak is indeed sensitive to it, as can be seen from the Figure X3 here below which uses a bandwidth of 2 deg. In the new version of the manuscript, we use a version with a bandwidth of 1.25 deg, which is the half of the grid used and it is approximately equal to the one obtained for the Silverman's rule of thumb. This is now specified in the figure's caption.

[Figure]

Figure X3: As Figure 3 from the manuscript, but with a 2deg bandwidth

However, we agree with the reviewer that a fourth peak is evident at TL799 around 50N. We however noticed that also in the original TL255-ctrl a "shoulder" at the same latitude is found also for ERA5. Even taking into consideration that we have only a single integration so that all the fine-scale discussion must be taken with a grain of salt, climate models are not new for having spurious peaks (Kwon et al. 2018). Overall, it is hard thus to conclude that the distribution does not converge also qualitatively, having a more correct distribution between the three peaks. A comment on the remaining biases at TL799 is now included in the text.

**28. L257 : please remove 'obviously'**
This has been removed.

**29. L261 : I do not fully agree with the conclusion. Especially over mainland Europe, increase in orography resolution worsens the bias. In addition, over the UK and Scandinavian regions, changes due to atm res. are comparable (and in cases larger) than the changes due to oro. res. Therefore, the statement is inarguably true only for the lower latitude Azores Atlantic basin.**
We agree with the reviewer that our initial statement was biased and too optimistic. In the new version of the manuscript we rephrased the discussion with a more balanced view, which is now confirmed by the statistical test included in Figure 5.

**30. L268 : Again, I do not fully agree with the conclusion. The changes in blocking duration are definitely comparable to the bias between T255 and ERA5, along with worsening of the bias over mainland Europe. While I support the conclusion over the Atlantic, concluding the same for mainland Europe is not so straightforward.**
Regarding this point we are not sure we understood the referee's comment. The change in blocking duration over mainland Europe is of the opposite sign than the bias between T255 and ERA5. Overall, the TL255 bias shows negative values reporting underestimated

blocking frequency, while increasing the orography resolution produces a widespread increase of blocking duration, albeit barely significant. It is true that in some instances, such as the Central Atlantic, the increase in duration goes in the wrong direction; but overall the orographic resolution shows a much more uniform improvement of blocking duration, which is confirmed by the figures of the PCI and RRI.

**31. L270 : The stationary patterns by changes in atmos. res and orog. res. are affected similarly (Figure 2b,c). Thus I think the final statement of the paragraph should be framed more carefully.**

We agree with the reviewer that this point was not clear: we wanted to refer to the changes in the stationary wave pattern structure over the North America continent, which is the one most affected by the changes in orography as can be shown in Figure 2b,c. This has serious implications on the jet stream structure and how the eddies propagate along the storm track, finally impacting the wave breaking and blocking activity. We rephrased the sentence in order to make it more close to this description.

**32. L275 : Is the bias significantly stronger than for the tuned T255 run?**

An initial clarification is required about the definition of "tuned" run: the TL255-ctrl integration, which is our baseline integration, is a tuned configuration, since the tuning performed by the EC-Earth consortium at this resolution was aimed at achieving a correct net surface radiative balance and surface temperature (Doescher et al. 2021). Those features are therefore valid also for our integration. Of course, removing the orographic parametrizations deteriorates the wind field, and has considerable implications on jet stream variability and blocking. Indeed, the impact of the SSO and TOFD parametrization is markedly strong, and widely significant as can be now appreciated by Figure SM1. The figure is now mentioned in the text and the whole paragraph about blocking has been rewritten.

**33. Figure 5 : Is the large red band over Eurasia in the bias (Figure 5a) explained exclusively by the Alps? (similar to the bias over western North America)**

We do not believe that local orography as Alps or Ural are capable of producing the large biases and consequent changes seen in Figure 5. Our interpretation is that eddy-mean flow interaction induced by increased/decreased drag from the Rocky Mountains leads to a weakening/strengthening Atlantic jet structure, reducing/increasing the number transient eddies which reach Europe and propagate further East over the Asian continent.

**34. L283 : ... orographic resolution. The changes to the ... I suggest a paragraph break here.**

Corrected.

**35. L286 : same terms (not shown)?**

Corrected.

**36. L295 : This is not totally clear to me, can you please explain why?**

At this line we are discussing the fact that the barotropic conversion term has a specific bias, which is shown in Figure 5g, which suggests that the eddy forcing strengthens the Atlantic jet in the equatorward side more than in ERA5 reanalysis, so that this supports the view of a southward displaced jet. We clarified this point in the new version of the manuscript.

**37. L317 : significantly for almost**
Corrected.

**38. L357 : I think it decreases by 1% at T511 and by 2% at T799 instead. Isn't it?**
Sorry for the mistake, we originally wrote the text for the tropical changes which are larger than the global ones. This is now corrected.

**39. L396 : ... might be considered responsible for the absence …**
This sentence has been rewritten.

**40. Figure 1(b) : please mention the maximum temperature difference associated with stratospheric cooling. The colorbar saturates early. It would also be great if the authors can mention in the Figure 4's caption the maximum value of the biases in Figure 4a as the colorbar saturates early for lower values. More generally, please increase the font size in all the figures, and wherever the colorbar saturates, provide the maximum value of the statistic.**
We changed the legend from linear to irregular in Figure 1, 2 and 4 in order to accommodate the different magnitude of changes, so that it is possible to extract the correct information from all the figures.

**41. How does the Jet Latitude Index (JLI) compare for the different models integrations over the Pacific? This is part of my broader concern, regarding how the wind changes due to flow blocking around the Rocky Mountains and Tibetan Plateau relate to changes in the Atlantic JLI.**
Overall, the Pacific jet variability is significantly different, having the jet much less meridional variability. Not surprisingly, a measure of the JLI over the Pacific (i.e. 150E-150W) shows a radically different scenario from the Atlantic. The jet is unimodal, as shown in Figure X4 here below.

[Figure]

Figure X4: As Figure 3, but for Central Pacific (150E-150W)

This agrees with what was discussed by the original Woollings et al. (2010) paper and seen also in models (Anstey et al. 2013). Concerning the impact of resolution and orography we see marginal changes, suggesting that the Tibetan Plateau has a less marked impact on the flow than the Rocky Mountains. Looking in detail, a marginal poleward displacement is seen when the atmospheric resolution is increased (see TL511-orog255 and TL799-orog255), and this is counterbalanced by the increase in orography that moves equatorward the JLI distribution. However, the overall result is a TL799-ctrl with a bias very similar to the one seen for the TL255-ctrl, characterized by a poleward displaced jet.

---

## Author Response (AR2)

**Replies to reviewer #1**

Line 47. Relies on, not in

Thanks for spotting, corrected.

Line 283. I don't understand what you mean by the responses being opposite and almost complementary. Suggest you rephrase, or just remove "almost complementary"

We removed "almost complementary" as suggested.

Line 320. I don't necessarily agree with the assessment that "most" of the improvement is due to the orographic resolution, but just looking at the graph is quite subjective. I would say, either come up with a way to quantify this (error at the peaks, or RMSE between ERA5 and experiment over particular latitude bands) to back up your statement of "most", or reduce the strength of this statement to "at least half", which I think is reasonable based on the figure.

We rephrased according to the reviewer comment, mentioning only "at least half" instead of "most"

Line 464-7. I think you mean Fig. 7e/f not, Fig 1e/f?

Correct, thanks for spotting it

**Line 485. Wording: the budget itself isn't necessarily larger, it's the radiative imbalance that is larger.**

We rephrased saying "which shows larger radiative imbalance".

Line 503. It is a key point that re-tuning isn't performed in the PRIMAVERA and HighResMIP projects, but the language here doesn't make that clear (it is unclear whether the 'procedure' in 'a procedure typical of...' is tuning the model, or not-re-tuning the model). Suggest to rephrase.

Line 505. "absence of improvement" seems to be overstating your results – there is improvement from the increase atmospheric resolution, it's just weaker than the improvement from orographic resolution increase. Suggested rephrase to "smaller improvement"..." relative to increase orographic resolution" or similar.

We rephrased the entire paragraph to accomodate for reviewers suggestions: "The set up of the presented experiments followed the typical approach of the initiatives aiming at assessing the impact of horizontal resolution increases in GCMs, as the PRIMAVERA H2020 project or the HighResMIP project: EC-Earth3 was tuned only once at the standard resolution (TL255) and no re-tuning was performed for different resolutions (TL511, TL799). However, increasing the atmospheric resolution caused large changes in the radiative budget at the TOA, suggesting that the lack of a proper re-tuning of the finer resolution configurations can be considered as the mechanism responsible for the minor improvements observed when the atmospheric resolution is increased."